# Towards Establishing Guaranteed Error for Learned Database Operations

**Sepanta Zeighami**[*]
UC Berkeley
`zeighami@berkeley.edu`

**Cyrus Shahabi**
University of Southern California
`shahabi@usc.edu`

## Abstract

Machine learning models have demonstrated substantial performance enhancements over non-learned alternatives in various fundamental data management operations, including indexing (locating items in an array), cardinality estimation (estimating the number of matching records in a database), and range-sum estimation (estimating aggregate attribute values for query-matched records). However, real-world systems frequently favor less efficient non-learned methods due to their ability to offer (worst-case) error guarantees — an aspect where learned approaches often fall short. The primary objective of these guarantees is to ensure system reliability, ensuring that the chosen approach consistently delivers the desired level of accuracy across all databases. In this paper, we embark on the first theoretical study of such guarantees for learned methods, presenting the necessary conditions for such guarantees to hold when using machine learning to perform indexing, cardinality estimation and range-sum estimation. Specifically, we present the first known lower bounds on the model size required to achieve the desired accuracy for these three key database operations. Our results bound the required model size for given average and worst-case errors in performing database operations, serving as the first theoretical guidelines governing how model size must change based on data size to be able to guarantee an accuracy level. More broadly, our established guarantees pave the way for the broader adoption and integration of learned models into real-world systems.

## 1 Introduction

Recent empirical results show that learned models perform many fundamental database operations (e.g., indexing, cardinality estimation) more efficiently than non-learned methods, providing significant speed-ups and space savings (Galakatos et al., 2019; Kraska et al., 2018; Ferragina & Vinciguerra, 2020; Zeighami et al., 2023; Kipf et al., 2018). Nevertheless, the lack of theoretical guarantees on their performance poses a significant hurdle to their practical deployment, especially since the non-learned alternatives often provide the required theoretical guarantees (Agarwal et al., 2013; Pătraşcu & Thorup, 2006; Hellerstein et al., 1997; Bayer & McCreight, 1970). Such guarantees are needed to ensure the reliability of the learned operations across all databases at deployment time, that is, to ensure consistent performance of the learned model on databases where the learned model had not been apriori evaluated. Thus, similar to existing worst-case bounds for non-learned methods, a guarantee is needed that a learned operation will achieve the desired accuracy level on all possible databases. Providing such a guarantee depends on how large the learned model is (e.g., number of parameters of a neural network), the desired accuracy level, and the size and dimensionality of the underlying databases. This paper takes the first step towards a theoretical understanding of the relationship between these factors for three key database operations, offering theoretical bounds on the required model size to achieve a desired accuracy on all possible databases of a certain size and dimensionality when using learned models to perform the operation.

Specifically, the three operations studied in this paper are (1) indexing: finding an item in an array, (2) cardinality estimation: estimating how many records in a database match a query, and (3) range-sum estimation: estimating the aggregate value of an attribute for the records that match a query. We focus on numerical datasets and consider axis-aligned range queries for cardinality and range-sum estimation (i.e., queries that ask for the intersection of ranges across dimensions). Typical

---

[*]This work was completed when the author was a PhD student at USC's Infolab

| Database Operation | Worst-Case Error | Average-Case Error (Uniform Dist.) | Average-Case Error (Arbitrary Dist.) |
|---|---|---|---|
| Indexing | $\frac{n}{2\epsilon+1}\log(1+\frac{(2\epsilon+1)u}{n})$ 
 THEOREM 1 | $(\sqrt{n}-2)\log(1+\frac{1}{2\epsilon})$ 
 THEOREM 2 | $(\sqrt{n}-2)\log(1+\frac{1}{2\epsilon})$ 
 THEOREM 5 |
| Cardinality Estimation | $\frac{n}{2\epsilon+1}\log(1+\frac{(2\epsilon+1)u^d}{n})$ 
 THEOREM 1 | $(\sqrt{n}-2)\log(1+\frac{\sqrt{n}^{d-1}}{4^{d(d+1)}\epsilon^d}-\frac{1}{\sqrt{n}})$ 
 THEOREM 3 | X 
 LEMMA 2 |
| Range-Sum Estimation | $\frac{n}{2\epsilon+1}\log(1+\frac{(2\epsilon+1)u^d}{n})$ 
 THEOREM 1 | $(\sqrt{n}-2)\log(1+\frac{\sqrt{n}^{d-1}}{4^{d(d+1)}\epsilon^d}-\frac{1}{\sqrt{n}})$ 
 COROLLARY 1 | X 
 LEMMA 2 |

Table 1: Our bounds on required model size in terms of data size, $n$, dimensionality, $d$, tolerable error, $\epsilon$, and domain size, $u$. Each column shows the result when $\epsilon$ is the tolerable error for the specified error scenario. X: No non-trivial bound possible. Base of $\log$ is 2.

learned approaches to the above database operations take a function approximation view of the operations. Let $f(q)$ be a function that takes a query, $q$, as an input, and outputs the answer to the query calculated from the database. For instance, in the case of cardinality estimation, $f(q)$ will be the number of records in the dataset that match the query $q$ (and $f(q)$ can be similarly defined for indexing and range-sum estimation). At training time, a model, $\hat{f}(q;\theta)$ (e.g., a neural network) is trained to approximate $f$. Training is done using supervised learning, where training labels are collected for different queries by performing the queries on the database using an existing method (e.g., for cardinality estimation, by iterating over the database and counting how many records match a query). At test time, the models are used to obtain estimates directly (e.g., by performing a forward pass of a neural network), providing $\hat{f}(q;\theta)$ as an estimate to the answer to a query $q$. For indexing, where the exact location of the query in the array is needed (not an estimated location returned by the model), a local search around the model estimate is performed to find the exact answer.

Such learned approaches are currently state-of-the-art, with experimental results showing significantly faster query time and lower storage space when using learned methods compared with non-learned methods for indexing (Kraska et al., 2018; Ferragina & Vinciguerra, 2020; Ding et al., 2020), cardinality estimation (Kipf et al., 2018; Negi et al., 2023) and range-sum estimation (Zeighami et al., 2023). Furthermore, recent results also show theoretical advantages to using learned models (Zeighami & Shahabi, 2023; Ferragina et al., 2020; Zeighami et al., 2023), most significantly, with Zeighami & Shahabi (2023) showing the existence of a learned index that can achieve expected query time of $O(\log\log n)$ under mild assumptions on the data distribution, asymptotically better than the traditional $O(\log n)$ of non-learned methods such as binary search. However, there has been no theoretical understanding of the required modeling choices, such as the required model size, for the learned approaches to provide an error guarantee across databases. Without any theoretical guidelines, design choices are made through empirical hyperparameter tuning, leading to choices with unknown performance guarantees at deployment time.

## 1.1 OUR RESULTS

In this paper, we present the first known bounds on the model size needed to achieve a desired accuracy when using machine learning to perform indexing, cardinality estimation and range-sum estimation. We provide bounds on the *required model size*, defined as the smallest possible size for a model to achieve error at most $\epsilon$ on all $d$-dimensional datasets of size $n$. We measure model size in terms of number of bits required to store a model (which translates to the number of model parameters by considering the storage precision for the parameters). We refer to $\epsilon$ as the *tolerable error parameter*, which denotes the maximum error that can be tolerated in the system. We thoroughly study the required model size in two different scenarios, namely when considering the worst-case and average-case error. That is, $\epsilon$ can be provided in terms of worst-case or average-case error across queries that can be tolerated for *all* databases (i.e., worst-case across databases). Table 1 summarizes our main results, which we further discuss considering the two error scenarios in turn.

First, suppose our goal is to answer *all possible queries with error at most $\epsilon$* across *all* $d$-dimensional datasets of size $n$. The results in the second column of Table 1, summarizing our Theorem 1 in Sec 3.1, provide a lower bound on the required model size to achieve this. For example, for indexing, to be able to guarantee error at most $\epsilon$ on all possible queries and datasets of size $n$, one must use a model whose size exceeds $\frac{n}{2\epsilon+1}\log(1+\frac{(2\epsilon+1)u}{n})$. Notably, the bounds depend on the domain size $u$, which is the number of possible values the records in the database can take, implicitly assuming a finite data domain. We show in Lemma 1 in Sec. 3.1 that this is necessary: no model with finite

size can answer queries with a bounded worst-case error on all possible datasets with infinite domain (this result is mostly of theoretical interest, since data stored in a computer always has finite domain).

In the second scenario, our goal is to answer *queries with average error* of at most $\epsilon$ on *all* $d$-dimensional datasets of size $n$. Assuming the queries are uniformly distributed, the third column in Table 1, summarizing Theorem 2, 3 and Corollary 1 in Sec. 3.2, presents our lower bounds on the required model size. Our bounds in this scenario show a weaker dependency on data size and tolerable error parameter compared with the worst-case error scenario, and as expected, suggest smaller required model size. Interestingly, bounds do not depend on the domain size and hold when the data domain is the set of real numbers, showing a significant difference between model size requirements when considering the two scenarios. Thus, our results formally show that robustness guarantees (i.e., guarantees on worst-case error) must come at the expense of larger model sizes.

Furthermore, the results in the last column of Table 1, summarizing our Theorem 5 and Lemma 2 in Sec. 3.3, show that we can extend our results to arbitrary query distribution (compared with uniform distribution) in the case of indexing without affecting the bounds. However, for cardinality and range-sum estimation, we show in Lemma 2 that when relaxing our assumption on data distribution, one can construct arbitrarily easy distribution to answer queries from, so that no non-trivial lower bound on the model size can be obtained (surprisingly, this is not possible for learned indexing).

Finally, not presented in Table 1, for average-case error, we complement our lower bounds on the required model size with corresponding upper bounds, showing tightness of our results. Theorem 2-5 show that our lower bounds are tight up to an $O(\sqrt{n})$ factor, asymptotically in data size.

## 1.2 DISCUSSION

In practice, model size is often set in practice to a fixed value or through hyperparamter tuning (Lu et al., 2021; Kraska et al., 2018; Kipf et al., 2018; Zeighami et al., 2023) without taking data size into account[1]. Our results show that model size indeed needs to depend on data size to be able to guarantee any fixed error. More specifically, for practical purposes, our results can be interpreted in two ways. In the first interpretation, given a model size and data size, our results provide a lower bound on the worst-case possible error. This bound shows what error can be guaranteed by a model of a certain size (and how bad the model can get) after it is deployed in practice. This is important, because datasets change in practice and our bound on error help quantify if a model of a given size can guarantee a desired accuracy level when the dataset changes. Experiments in Sec. 4 illustrate that this bound on error is meaningful, showing that models achieve error values close to what the bound suggests. In the second interpretation, our results provide a lower bound on the required model size to achieve a desired accuracy level across datasets. This shows how large the model needs to be, to guarantee the desired accuracy, and has significant implications for resource management in database systems. For instance, it helps a cloud service provider decide how much resources it needs to allocate for models to be able to guarantee an accuracy level across all its database instances.

Overall, our results are information theoretic, showing that it is not possible for *any* model to contain enough information to answer queries on all datasets accurately if they contain less than the specified number of bits. The bounds are obtained by considering the parameters of a model as a data representation, and showing bounds on the required size of any data representation to achieve a desired accuracy when performing the specific operations. Our proofs provide a novel exploration of the function approximation view of database operations, connecting combinatorial properties of datasets with function approximation concepts. Specifically, we prove novel bounds on packing and metric entropy of the metric space of database query functions to prove the bounds, which are particularly challenging to obtain for the average-case error. In Sec.A, we discuss various possible extensions of our results to queries with joins, other aggregation function such as min/max/avg. and other error metrics not considered in this paper.

## 2 PRELIMINARIES

**Setup**. We are given a dataset, $\boldsymbol{D} \in \mathcal{D}^{n \times d}$, i.e., a dataset consisting of $n$ records and in $d$ dimensions with each attribute in the *data domain* $\mathcal{D}$, where $n$ and $d$ are integers greater than or equal to 1. Unless otherwise stated, we assume $\mathcal{D} = [0, 1]$ so that $\boldsymbol{D} \in [0, 1]^{n \times d}$ (attributes can be scaled to $[0, 1]$ if they fall outside the range). We use $\boldsymbol{D_i}$ to refer to the $i$-th record of the dataset (which is a $d$-dimensional vector) and $D_{i,j}$ to refer to the $j$-th element of $\boldsymbol{D_i}$. If $d = 1$ (i.e., $\boldsymbol{D}$ is 1-dimensional), then $D_i$ is the $i$-th element of $\boldsymbol{D}$ (and is not a vector). We study the following database operations.

---

[1]At least explicitly, as hyperparameter tuning can implicitly account for data size

*Indexing.* The goal is to find an item in a sorted array. Formally, consider a 1-dimensional sorted dataset $D$ (i.e., a sorted 1-dimensional array). Given a query $q \in [0, 1]$, return the index $i^* = \sum_{i=1}^{n} I_{D_i \leq q}$, where $I$ is the indicator function. $i^*$ is the index of the largest element no greater than $q$ and is 0 if no such element exists. Furthermore, if $q \in D$, $q$ will be at index $i^* + 1$. $i^*$ is referred to as the *rank* of $q$. Define the *rank function* of the dataset $D$ as $r_D(q) = \sum_{i=1}^{n} I_{D_i \leq q}$, which takes a query as an input and outputs its rank. We have $Q_r = [0, 1]$ as the domain of the rank function.

*Cardinality Estimation.* Used mainly for query optimization, the goal is to find how many records in the dataset match a range query, where the query specifies lower and upper bound conditions on the values of each attribute. Formally, consider a $d$-dimensional dataset. A query predicate $q = (c_1, ..., c_d, r_1, ..., r_d)$, specifics the condition that the $i$-th attribute is in the interval $[c_i, c_i + r_i]$. Define $\mathcal{I}_{p,q}$ as an indicator function equal to one if a $d$-dimensional point $p = (p_1, ..., p_d)$ matches a query predicate $q = (c_1, ..., c_d, r_1, ..., r_d)$, that is, if $c_j \leq p_j \leq c_j + r_j, \forall j \in [d]$ ($[k]$ is defined as $[k] = \{1, ..., k\}$ for integers $k$). Then, the answer to a cardinality estimation query is the number of points in $D$ that match the query $q$, i.e., $c_D(q) = \sum_{i \in [n]} \mathcal{I}_{D_i, q}$. We refer to $c_D$ as the *cardinality function* of the dataset $D$, which take a query as an input and outputs the cardinality of the query. We define $Q_c = \{r_j \in [0, 1], c_j \in [-r_j, 1 - r_j], \ j \in [d]\}$, where the definition ensures $c_j + r_j \in [0, 1]$ to avoid asking queries outside of the data domain.

*Range-Sum Estimation.* The goal is to calculate the aggregate value of an attribute for the records that match a query. Formally, consider a $(d + 1)$-dimensional dataset $D$ and a query $q = (c_1, ..., c_d, r_1, ..., r_d)$, where $q$, similar to the case of cardinality estimation, defines lower and upper bounds on the data points. The goal is to return the total value of the $(d + 1)$-th attributes of the points in $D$ that match the query $q$, i.e., $s_D(q) = \sum_{i \in [n]} \mathcal{I}_{D_i, q} D_{i, d+1}$. Here, for simplicity, we overload the notation and use $\mathcal{I}_{p,q}$, when the dimensionality of the query and predicate doesn't match to be defined as $c_j \leq p_j \leq c_j + r_j, \forall j \in [\min\{d, d'\}]$, where $d$ is dimensionality of the point $p$ and $d'$ is the dimensionality of the predicate $q$. $s_D$ is called the *range-sum function* of the dataset $D$, which takes a query as an input and outputs the range-sum of the query. We define the range-sum function domain $Q_s$ to be the same as $Q_c$.

We use the term *query function* to collectively refer to the rank, cardinality and range-sum functions, and use the notation $f_D \in \{r_D, s_D, c_D\}$ to refer to all the three functions, $r_D$, $c_D$ and $s_D$ (for instance, $f_D \geq 0$ is equivalent to the three independent statements that $r_D \geq 0$, $c_D \geq 0$ and $s_D \geq 0$). We drop the dependence on $D$ if it is clear from context and simply use $f(q)$. We also use $Q_f$ to refer to $Q_s$, $Q_c$ and $Q_r$ for $f \in \{r, c, s\}$.

For cardinality and range-sum estimation, often only an estimate of the query result is needed, because many applications (e.g., query optimization and data analytics) prefer a fast estimate over a slow but exact answer. For indexing, although exact answers are needed to locate an element in an array, one can do so through approximation. First, an estimate of the rank function is obtained, and then, a local search of the array around the provided estimate (e.g., using exponential or binary search) leads to the exact result. Thus, in all cases, approximating the query function with a desired accuracy is the main component in answering the query, which is the focus of the rest of this paper.

**Learned Database Operations**. Learned database operations use machine learning to approximate the database operations as follows. First, during training, a function approximator, $\hat{f}(.; \theta)$ is learned to approximate the function $f$, for $f \in \{r, s, c\}$. This is typically done through supervised learning (although unsupervised approaches are also possible), where for different queries sampled from $Q_f$, the operations are performed on the database to find the ground-truth answer, and the models are optimized through a mean squared loss. Subsequently, at test time, for a test query $q$, $\hat{f}(q; \theta)$ is used as an estimate of the query answer, which is obtained by performing a forward pass of the model. In practice, models used have a much fewer number of parameters than the data size, so that the models don't memorize the data but rather utilize patterns in query answers to perform the operations, leading to the practical gains in query answering.

This procedure can be formally specified as follows (both for supervised and unsupervised approaches). First, a function $\rho(D)$, takes the dataset as an input and generates model parameters $\theta$ (e.g., through training with gradient descent). Then, to answer a query $q$ at test time, the function $\hat{f}(q; \theta)$ takes both the model parameters and the query as input and provides the final query answer (i.e., $\hat{f}$ specifies the model forward pass). From this perspective, the model parameters $\theta$ is

a representation of the dataset $\boldsymbol{D}$ and the function $\hat{f}$ only uses this representation to answer queries, without accessing the data itself. We call $\rho$ the *training function* and $\hat{f}$ the *inference function*.

**The Model Size Problem**. An important question is how large the model needs to be to be able to achieve error at most $\epsilon$ on datasets of size $n$. We quantify the model size in terms of the number of bits needed to store the parameters $\theta$. The required model size is formalized as below.

**Definition 1** (Required Model Size). *Let $f \in \{r, c, s\}$ and consider an error norm $\|.\|$ for functions approximating $f$, and a data domain $\mathcal{D}$. Let $\mathcal{F}_\sigma$ be the set of all possible training and inference function pairs, where the training function generates a parameter set of size at most size $\sigma$ bits. Let $\Sigma$ be the smallest $\sigma$ so that there exists $(\rho, \hat{f}) \in \mathcal{F}_\sigma$ such that $\|\hat{f}(.; \rho(\boldsymbol{D})) - f_D\| \leq \epsilon$ for all $\boldsymbol{D} \in \mathcal{D}^{n \times d}$. We call $\Sigma$ the required model size to achieve $\|.\|$-error of at most $\epsilon$ in the worst-case across all $d$-dimensional datasets of size $n$.*

$\Sigma$ is the size of the parameter set passed from the training function to the inference function. Thus, in the above formulation, the training/inference functions can be arbitrarily complex. The goal of this paper is to present lower bounds on $\Sigma$ in terms of $n$ and $\epsilon$, and depending on the error norm $\|.\|$, which is an important factor impacting the lower bounds. One expects that larger models are needed if the worst-case error over all queries is considered, compared with the average error. Specifically, for $f \in \{r, s, c\}$, we consider the 1-norm error of approximating $f$ with $\hat{f}$ as $\|f - \hat{f}\|_1 = \int_{Q_f} |f - \hat{f}|$, the $\infty$-norm error as $\|f - \hat{f}\|_\infty = \sup_{q \in Q_f} |f(q) - \hat{f}(q)|$ and the $\mu$-norm $\|f - \hat{f}\|_\mu = \int_{Q_f} |f - \hat{f}| d\mu$ where $\mu$ is a probability measure over $Q_f$. $\infty$-norm is also called the worst-case error and $\mu$-norm error is referred to as the average error with arbitrarily distributed queries and 1-norm is referred to as the average error with uniformly distributed queries (note that the volume of query space is 1 for all the function domains, so that 1-norm indeed corresponds to uniform distribution).

# 3 LOWER BOUNDS ON MODEL SIZE FOR DATABASE OPERATIONS

We present lower bounds on the required model size to be able to provide worst-case and average-case error guarantees. For all cases, our results provide lower bounds for achieving error $\epsilon$ on *all* datasets. In other words, we show that if the model size is smaller than a specific threshold, then there exists a dataset such that the error is larger than $\epsilon$. As such, our bounds consider the worst-case error across datasets, while considering either average or worst-case error across queries. We first present our results considering the worst-case error in Sec. 3.1, then present results considering average-case error in Secs. 3.2 and 3.3 for uniform and arbitrary query distributions, respectively. Proof of our results are presented in Sec. B.

## 3.1 BOUNDS CONSIDERING WORST-CASE ERROR

We present our results when considering the worst-case error, or $\infty$-norm in approximation.

First, for the purpose of the following theorem, suppose the datasets are discretized at the unit $\frac{1}{u}$, that is datasets are from the set $\mathcal{D}_u = \{\frac{i}{u}, u \in [u]\}^{n \times d}$ (this reduces the data domain from the set of real numbers in $[0, 1]$ to multiples of $\frac{1}{u}$ in $[0, 1]$). Define $\Sigma_f^\infty$ for $f \in \{r, c, s\}$, as the required model size to answer queries to $\infty$-norm error at most $\epsilon$ for all datasets in $\mathcal{D}_u$. For instance, $\Sigma_r^\infty$ is the smallest possible model size to be able to approximate rank function with $\infty$-norm at most $\epsilon$ on all possible datasets from the data domain $\mathcal{D}_u$.

**Theorem 1.** *For any error $1 \leq \epsilon < \frac{n}{2}$,*

   (i) *For the case of learned indexing, we have that $\Sigma_r^\infty \geq \frac{n}{2\epsilon+1} \log(1 + \frac{(2\epsilon+1)u}{n})$,*

   (ii) *For the case of learned cardinality estimation, we have that $\Sigma_c^\infty \geq \frac{n}{2\epsilon+1} \log(1 + \frac{u^d(2\epsilon+1)}{n})$, and*

   (iii) *For the case of learned range-sum estimation, we have that $\Sigma_s^\infty \geq \frac{n}{2\epsilon+1} \log(1 + \frac{u^d(2\epsilon+1)}{n})$.*

The theorem provides lower bounds on the required model size to be able to perform database operations with a desired accuracy. For instance, for the case of learned indexing, the theorem states that the model size must be larger than $\frac{n}{2\epsilon+1} \log(1 + \frac{(2\epsilon+1)u}{n})$ to be able to guarantee $\infty$-norm error $\epsilon$ on all datasets, or alternatively, that if the model size is less than $\frac{n}{2\epsilon+1} \log(1 + \frac{(2\epsilon+1)u}{n})$, then for any model approximating the rank function, there exists a database where the model's $\infty$-norm error is more than $\epsilon$. We see that the required model size is close to linearly dependent on data size and

dimensionality, while inversely correlated with the tolerable error paramter. Besides dependence on data size, dimensionality and error, the bound shows a dependence on $u$, the domain size. An interesting question, then, is whether similar bounds on model size will hold if the data domain is not finite. The next lemma shows that the answer is no.

**Lemma 1.** *When $\epsilon < \frac{n}{2}$ and the data domain $\mathcal{D} = [0,1]$, for any finite size $\sigma$, and any training/inference function pair $(\rho, \hat{f}) \in \mathcal{F}_\sigma$, there exists a dataset $\boldsymbol{D} \in [0,1]^n$ such that $\|\hat{f}(.; \rho(\boldsymbol{D})) - r_D\|_\infty > \epsilon$.*

We remark that Lemma 1 may not be surprising. Storing real numbers requires infinitely many bits, and, although we are interested in the size of the model required to answer queries (and not the space required to store the data), one might expect the model size should be similar to the space required to store that data. Lemma 1 shows this to be true in the case of worst-case error. However, perhaps more surprisingly, the remainder of our results in the next sections show this is not true when considering the average-case error. As such, we consider the case where $\boldsymbol{D} \in [0,1]^{n \times d}$ for the remainder of this paper.

### 3.2 BOUNDS CONSIDERING AVERAGE-CASE ERROR WITH UNIFORM DISTRIBUTION

In this section, our results provide the average-case error assuming uniform data distribution, or 1-norm approximation error. Average-case error corresponds with the expected performance of the system, another important measure needed for real-world deployments. A bound on 1-norm error across all possible databases provides a performance guarantee for all possible databases. Our results in this section show how large the model needs to be to provide such guarantees.

#### 3.2.1 LEARNED INDEXING

We first present our result showing a lower bound on the required model size for learned indexing.

**Theorem 2.** *Let $\Sigma_r^1$ be the required model size to achieve 1-norm error of at most $\epsilon$ on datasets of size $n$ when approximating the rank function.*

*(i) For any $0 < \epsilon \le \frac{\sqrt{n}}{2}$, we have that $\Sigma_r^1 \ge (\sqrt{n} - 2) \log(1 + \frac{1}{2\epsilon} - \frac{1}{\sqrt{n}})$.*

*(ii) For any $0 < \epsilon \le n$, we have that $\Sigma_r^1 \le n \log\left(e + \frac{e}{\epsilon} + \frac{e}{n}\right)$.*

Part (i) of the theorem states that, if a model whose size is less than $(\sqrt{n} - 2) \log(1 + \frac{1}{2\epsilon} - \frac{1}{\sqrt{n}})$ bits is used to approximate the rank function, then there will exist a dataset of size $n$ for which the model results in error larger than $\epsilon$. As expected, the required model size increases both as data size increases, and as error threshold decreases. Furthermore, for a constant error $\epsilon$, the results shows that the require model size is $\Omega(\sqrt{n})$, providing the first known results showing the increase of model size with data size has to be at least in the order of $\sqrt{n}$.

Part (ii) of the theorem shows that the asymptotic dependence on $n$ in the lower bound is tight up to a $\sqrt{n}$ factor and to achieve a desired accuracy, model size does not need to increase more than linearly in data size. Overall, Theorem 2 shows that for a constant error, $\Sigma_r^1$ is $\Omega(\sqrt{n})$ and $O(n)$. The proof of part (ii) constructs a model that achieves the bound. The model can be seen as a nearest neighbor encoder, modeling a dataset based on its nearest neighbor in a constructed set of datasets.

Observe that this, and the rest of our results considering average case error do not depend on the domain size (as Theorem 1 did). Thus, a fundamental difference between answering queries accurately in the worst case, compared with average case is that in the first scenario the lower bounds depend on the discretization unit, while in the second scenario, model size does not depend on the discretization unit (i.e., Theorems 2-4). Furthermore, our results show that the lower bound in the case of the worst-case error has a stronger dependence on the tolerable error parameter, compared with when average error is considered.

#### 3.2.2 LEARNED CARDINALITY ESTIMATION

Next, we present an analog of Theorem 2 for the case of cardinality estiamtion.

**Theorem 3.** *Let $\Sigma_c^1$ be the required model size to achieve 1-norm error at most $\epsilon$ on $d$-dimensional datasets of size $n$ when approximating the cardinality function.*

*(i) For any $0 < \epsilon \le \frac{\sqrt{n}}{4^d}$, we have that $\Sigma_c^1 \ge (\sqrt{n} - 2) \log(1 + \frac{\sqrt{n}^{d-1}}{4^{d(d+1)}\epsilon^d} - \frac{1}{\sqrt{n}})$.*

*(ii) For any $0 < \epsilon \le n$, we have that $\Sigma_c^1 \le n \log\left(\frac{e2^d(d+1)^d n^{d-1}}{\epsilon^d} + e - \frac{e}{n}\right)$.*

The above theorem shows that, in the case of cardinality estimation, bounds of a similar form to the case of indexing hold, however, the bounds now also depend on data dimensionality. We see that, asymptotically in data size, $\Sigma_c^1$ is $\Omega(d\sqrt{n}\log(\frac{\sqrt{n}}{4^d\epsilon}))$ and $O(dn\log\frac{2dn}{\epsilon})$, where we see a close to linear required dependency on dimensionality while there is also an additional logarithmic dependency on $n$ compared with the case of indexing.

### 3.2.3 RANGE-SUM ESTIMATION

Finally, we extend our results to range-sum estimation. For the discussion in this section, let $\Sigma_s^1$ be the required model size to achieve 1-norm error at most $\epsilon$ on $(d+1)$-dimensional datasets of size $n$ when approximating the range-sum function. Recall that we consider $d+1$ dimensional datasets here, where query predicates apply to the first $d$ dimensions and the query answers are the aggregation of the $(d+1)$-th dimension, as defined in Sec. 2.

To prove a lower bound on $\Sigma_s^1$, observe that range-sum estimation can be seen as a generalization of cardinality estimation. Specifically, answering range-sum queries on a $d+1$-dimensional dataset, where the $d+1$-th attribute of all the records is set to 1, is equivalent to answering cardinality estimation queries on the $d$-dimensional dataset consisting only of the first $d$ dimensions of the original dataset. Thus, if a model with size less than $\Sigma_s^1$ is able to answer range-sum queries on all datasets with error at most $\epsilon$, then it can also answer cardinality estimation queries with error at most $\epsilon$. This means the lower bound on model size from Thoerem 3 (i) translates to range-sum estimation as well. Thus, we have the following result as a corollary to Thoerem 3.

**Corollary 1.** *For any $0 < \epsilon \leq \frac{\sqrt{n}}{4^d}$, we have that $\Sigma_s^1 \geq (\sqrt{n} - 2)\log(1 + \frac{\sqrt{n}^{d-1}}{4^{d(d+1)}\epsilon^d} - \frac{1}{\sqrt{n}})$.*

Next, we show that an upper bound very similar to Theorem 3 (ii) on the required model size also holds for range-sum estimation.

**Theorem 4.** *For any $0 < \epsilon \leq n$, we have that $\Sigma_s^1 \leq n\log(e(\frac{2(d+2)}{\epsilon})^{d+1}n^d + e - \frac{e}{n})$.*

Observe that the upper bound on $\Sigma_s^1$ is similar to $\Sigma_c^1$, but slightly larger, showing a stronger dependence on dimensionality in the case of $\Sigma_s^1$. This reflects the discussion above, that range-sum estimation is a generalization of cardinality estimation. Indeed, the proof of Theorem 4 is a generalization of the proof of Theorem 3 (ii).

### 3.3 BOUNDS CONSIDERING AVERAGE-CASE ERROR WITH ARBITRARY DISTRIBUTION

Next, we discuss extending the results in Sec. 3.2 to an arbitrary query distribution. The following theorem shows that this generalization does not impact the bounds in the case of indexing, i.e., the theorem below shows that the same bounds as in Theorem 2 also hold when considering $\mu$-norm.

**Theorem 5.** *Let $\Sigma_r^\mu$ be the required model size to achieve $\mu$-norm error at most $\epsilon$ on datasets of size $n$ when approximating the rank function, for any continuous probability measure $\mu$ over $[0, 1]$.*

   *(i) For any $0 < \epsilon \leq \frac{\sqrt{n}}{2}$, we have that $\Sigma_r^\mu \geq (\sqrt{n} - 2)\log(1 + \frac{1}{2\epsilon} - \frac{1}{\sqrt{n}})$.*

   *(ii) For any $0 < \epsilon \leq n$, we have that $\Sigma_r^\mu \leq n\log(e + \frac{e}{\epsilon} + \frac{e}{n})$.*

However, the next lemma shows that the lower bounds do not hold for arbitrary distributions in the case of cardinality and range sum estimation.

**Lemma 2.** *For $f \in \{c, s\}$, there exists a query distribution, $\mu$, such that for any error parameter $\epsilon > 0$, we have $\|f_D - f_{D'}\|_\mu \leq \epsilon$ for all $D, D' \in [0, 1]^{n \times d}$.*

The above lemma shows that one can construct a distribution for which the $\mu$-norm difference between all datasets is arbitrarily small. As a result, one can answer queries independently of the observed dataset, and therefore the required model size for achieving any error is 0. The proof of Lemma 2 creates a data distribution consisting only of queries with small ranges, so that the answer to most queries is zero or close to zero for any dataset. Thus, comparing Lemma 2 with Theorem 5, we see that queries having both a lower and upper bound on the attributes leads to a different theoretical characteristic for cardinality and range-sum estimation compared with indexing.

## 4 EMPIRICAL RESULTS

We present experiments comparing our bounds with the error obtained by training different models on datasets sampled from different distributions. We train a linear model, and two neural networks with a single hidden layer where the two neural networks have 10 and 50 model parameters, respectively referred to as NN-S1 and NN-S2. Small neural networks and linear models are common

Figure 1: Theoretical Bounds in Practice

modeling choices for learned database operations (Kraska et al., 2018; Ferragina & Vinciguerra, 2020; Kipf et al., 2018; Zeighami et al., 2023). We also present results for using a random samples as a non-learned baseline, referred to as Sample. For direct comparison, the number of samples are set so that Sample takes the same space as the linear model. We consider 1-dimensional datasets sampled from uniform and 2-component Gaussian mixture model distributions.

Recall that our theoretical results provide bounds of the form $\Sigma > g(n, \epsilon, d)$, for some function $g$ specified in our theorems. Given a model size, $\sigma$, data size, $n$ and dimensionality $d$, define $\epsilon^*$ as the largest $\epsilon$ such that $\sigma \leq g(n, \epsilon, d)$ holds. For model size $\sigma$, this implies that for any model, there exists a $d$-dimensional dataset of size $n$ where the error of the model is at least $\epsilon^*$. Thus, an interpretation of our theoretical results is that given a model size, $\sigma$, data size, $n$ and dimensionality, $d$, our results provide *a lower bound on the worst-case error* across all $d$-dimensional datasets of size $n$ for any model of size $s$, where this lower bound is equal to $\epsilon^*$ as defined above. Our experiments present results following this view of our theoretical bounds.

Our experimental results are presented in Fig. 1. The color of the lines/points in the figure corresponds to the specific models, and the points in the figures show either the maximum or average error across queries, observed after training the specific models on datasets sampled from either GMM or uniform distributions. The solid lines in the figures plot the value of $\epsilon^*$, i.e., lower bound on the worst-case error across datasets, for different model and data sizes. The results are presented for indexing and cardinality estimation and under 1-norm and $\infty$-norm errors. Our bounds on range-sum estimation are similar to cardinality estimation, and are thus omitted. Note that the error is often (much) larger than 1, since the error is based on the absolute difference between model prediction and true query answers. The true query answers can be large, and their values scale with data size, so that the absolute error of the models is also large and increases with data size. We perform no normalization of the error values, to allow direct comparison with our theoretical results.

First, consider our results on the worst-case error, shown in Figs. 1 (a) and (b). As expected, the observed error of different trained models increases with data size, with the models achieving lower error on uniform distribution compared with a GMM. Furthermore, our theoretical bounds on the model size lie close to the error of the models on GMMs, showing that the bounds are meaningful. In fact, in the case of the linear model, all the observed errors lie below the theoretical lower bound. This implies that, based on our theoretical result, there exists some dataset (on which the models haven't been evaluated in this experiment) whose error lies on or above the theoretical lower bound. This shows a practical benefits of our bound: one can obtain a bound on the error of the model on all possible databases, and beyond the datasets on which the model has been empirically evaluated.

Next, consider our results on the average-case error, shown in Figs. 1 (c) and (d). Compared with the worst-case scenario, we see that the results show a large gap between error of the models and our theoretical bounds, especially so for larger model sizes. Our tightness results in Sec. 3.2, not plotted here, theoretically quantify how large this gap can be. We also note tha, for both worst-case and average-case scenarios, the gap between the observed error and our lower bounds on large models does not necessarily imply that our bounds are looser for larger model sizes. Such a gap can also be due to the models used in practice being wasteful of their storage space for larger model sizes. Our results support the latter hypothesis, since we observe marginal improvement in accuracy as model size increases across both distributions. This is also supported by observations that sparse neural networks can achieve similar accuracy as non-sparse models while using much less space (e.g., Frankle & Carbin (2019)), hinting at the suboptimality of fully connected networks.

Finally, Fig. 1 shows that sampling performs worse than learned models for uniform distribution while it performs similarly for GMMs (Sample should be compared with Linear as they both have the same size). The latter can be because a linear model is not a good modeling choice for GMMs.

Nonetheless, our theoretical bounds suggest (see Sec. 5 for theoretical comparison) the gap between learned models and sampling will grow as dimensionality increases ($d = 1$ in our experiments).

## 5 RELATED WORK

A large and growing body of work has focused on using machine learning to speed up database operations, among them, learned indexing (Galakatos et al., 2019; Kraska et al., 2018; Ferragina & Vinciguerra, 2020; Ding et al., 2020), learned cardinality estimation (Kipf et al., 2018; Wu & Cong, 2021; Hu et al., 2022; Yang et al., 2019; 2020; Lu et al., 2021; Negi et al., 2021) and learned range-sum estimation (Zeighami et al., 2023; Hilprecht et al., 2019; Ma & Triantafillou, 2019). Most existing work focus on improving modeling choices, with various modeling choices such as neural networks (Zeighami et al., 2023; Kipf et al., 2018; Kraska et al., 2018), piece-wise linear approximation (Ferragina & Vinciguerra, 2020), sum-product networks (Hilprecht et al., 2019) and density estimators (Ma & Triantafillou, 2019). In all existing work, modeling choices are based on empirical observations and hyperparametr tuning, with no theoretical understanding of how the model size should be set for a dataset to achieve a desired accuracy.

On the theory side, existing results in learned database theory show existence of models that achieve a desired accuracy (Zeighami et al., 2023; Zeighami & Shahabi, 2023; Ferragina et al., 2020), showing bounds on the performance of specifically constructed models to perform database operation. Among them, Zeighami & Shahabi (2023) shows that a learned model can perform indexing in $O(\log \log n)$ expected query time (i.e., better than the traditional $O(\log n)$). Our results complement such work, showing a lower bound on the required size for any modeling approach to perform the database operations. We note that the bounds in Zeighami et al. (2023); Zeighami & Shahabi (2023); Ferragina et al. (2020) either hold on expectation or with a certain probability, while our bounds are non-probabilistic and consider the worst-case across all datasets. Orthogonal to our work, Hu et al. (2022); Agarwala et al. (2021) study the number of training samples needed to achieve a desired accuracy for different database operations.

More broadly, non-learned data models, such as samples, histograms, sketches, etc. (see e.g. Cormode et al. (2012); Cormode & Yi (2020); Shekelyan et al. (2017)) are also used to estimate query answers. We discuss existing lower bounds which is the focus of our paper, and refer the reader to Cormode et al. (2012); Cormode & Yi (2020) for a complete treatment of non-learned methods. One approach is using $\epsilon$-approximations, where a subset of the dataset is selected (through random sampling or deterministically) and used for query answering. Wei & Yi (2018); Matoušek & Nikolov (2015) show that for cardinality estimation, the size of such a subset has to be at least $\Omega(\frac{n}{\epsilon} \log^{d-1}(\frac{n}{\epsilon}))$ to answer queries with worst-case error at most $\epsilon$. Using the fact that VC dimension of orthogonal range queries is $2d$ (Shalev-Shwartz & Ben-David, 2014), we have that random sampling uses $O((\frac{n}{\epsilon})^2(d + \log \delta))$ samples to provide error at most $\epsilon$ with probability $\delta$ Mustafa & Varadarajan (2017), while Phillips (2008) provides a deterministic method using a subset of size $O(\frac{n}{\epsilon} \log^{2d}(\frac{n}{\epsilon}) \operatorname{polylog}(\log(\frac{n}{\epsilon})))$. Compared to our bound in Theorem 1, we observe that $\epsilon$-approximations can be much less space-efficient compared with other modeling choices in high dimensions. This is because $\epsilon$-approximations correspond to a restricted class of models, where the model of the data is simply a subset of the data. Relaxing this restriction, Wei & Yi (2018) considers a special case of our Theorem 1 (ii), where they provide a lower bound of $\frac{n}{\epsilon}(\log^2(\frac{n}{\epsilon}) + \log(n))$ on the required size when answering cardinality estimation queries in *two* dimensions and when $u = n$ (recall that $u$ is the discretization factor in Sec. 3.1). The lower bound is tighter than our bound in Theorem 1 (ii), but unlike our bound that applies to arbitrary dimensionsionality and granularity, is only applicable to the specialized case of $d = 2$ and $u = n$. In this setting, Wei & Yi (2018) also present a non-learned data structure that matches the lower bound. Orthogonal to our work, other lower bounds have been presented in the streaming setting (Cormode & Yi, 2020; Suri et al., 2004).

## 6 CONCLUSION

We presented the first known lower bounds on the required model size to achieve a desired accuracy when using machine learning to perform indexing, cardinality estimation and range-sum estimation. We studied the required model size when considering average-case and worst-case error scenarios, showing how the model size needs to change based on accuracy, data size and data dimensionality. Our results highlight differences in model size requirements when considering average-case and worst-case error scenarios and when performing different database operations. Our theoretical results provide necessary conditions for ensuring reliability of learned models when performing database operations. Future work includes providing tighter bounds for average-case error, bounds based on data characteristics such as data distribution, and studying other database operations.

ACKNOWLEDGMENTS

This research has been funded by NSF grant IIS-2128661 and NIH grant 5R01LM014026. Opinions, findings, conclusions, or recommendations expressed in this material are those of the author(s) and do not necessarily reflect the views of any sponsors, such as NSF.

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

## A  DISCUSSION

**Bounding** $\log_2$ **error**. Recall that our results consider the absolute error of prediction. In the case of indexing, one is often interested in $\log_2$ of the error, since that's the runtime of the binary search performed to find the true element after obtaining an estimate from the model. Our worst-case absolute error bound directly translates to worst-case $\log_2$ error bound (this is because $\log$ is an increasing function). That is, there exists a dataset such that worst-case absolute error is $\epsilon$ if and only if there exists a dataset such that worst-case $\log_2$ error is $\log_2 \epsilon$. Thus, to ensure $\log_2$ error is at most an error parameter $\tau$, we can directly set $\epsilon = 2^\tau$ in the bound presented in Theorem 1 to obtain the bound on required model size. Regarding the average-case error, the situation is slightly more complex. Using Jensen's inequality, we have that the average $\log_2$ error can be smaller than $\log_2$ of the average error. This implies that, to obtain average $\log_2$ error of $\tau$ the required model size may indeed be smaller than the bounds in Corollary 1 and part (i) of Theorems 2, 3, 5 would suggest if we set $\epsilon = 2^\tau$. Nonetheless, our upper bounds on the required model size (i.e., Theorem 4, Lemma 2 and part (ii) of Theorems 2, 3, 5) still apply by setting $\epsilon = 2^\tau$.

**Cardinality Estimation for Joins**. Cardinality estimation is often used to estimate cardinality of joins, which is important for query optimization. Our bounds present lower bounds on cardinality estimation on a single table. Note that a naive extension of our bounds to the cardinality estimation for joins is to apply the bound to the join of tables. That is, if two tables have respectively $d_1$ and $d_2$ dimensions and if their join consists of $n_J$ elements, then we can apply our bounds in Table 1 with $d = d_1 + d_2$ and $n = n_J$ to obtain a lower bound on the required model size for estimating the cardinality of the join. However, we expect such an approach to overestimate the required model size, as it does not consider the join relationship between the two tables. For instance, $n_J \times d$ may be much larger than $n_1 \times d_1 + n_2 \times d_2$, because of duplicate records created due to the join operations. Considering the join relationship, one may be able to provide bounds that depend on the original table sizes and not the join size.

**Other Aggregations**. Our results consider count and sum aggregations. Although we expect similar proof techniques as what was used to apply to other aggregations such as min/max/avg, we note that studying worst-case bounds for min/max/avg may not be very informative. Intuitively, this is because, for such aggregations, one can create arbitrary difficult queries that require the model to memorize all the data points. For instance, consider datasets where the $(d + 1)$-th dimension, where the aggregation is applied, only takes 0 or 1 values, while the other dimensions can take any values in a domain of size $u$. Now avg/min/max for any range query will have an answer between 0 and 1 (for min/max the answer be exactly 0 or 1). However, unless the model memorizes all the points exactly (which requires a model size close to the data size), its worst-case error can be up to 0.5. This is because queries with a very small range can be constructed that match exactly one point in the database, and unless the model knows the exact value of all the points, it will not be able to provide a correct answer for all such queries. Note that the error of 0.5 is very large for avg/min/max queries, and a model that always predicts 0.5 also obtains worst-case error 0.5. This is not the case for sum/count aggregations whose query answers range between 0 to $n$, and a model with an absolute error of 0.5 can be considered a very good model for most queries when answering sum/count queries.

To summarize, worst-case errors of min/avg/max queries are disproportionately affected by smaller ranges, while smaller ranges have a limited impact on the worst-case error of count/sum queries. As such we expect that for the case of min/max/avg queries, one needs to study the error for a special set of queries (e.g., queries with a lower bound on their range, as done in Zeighami et al. (2023), which also makes similar observations) to be able to obtain meaningful bounds. We leave such a study to future work.

## B  PROOFS

### B.1  INTUITION

Our proofs study the characteristics of the space of the query functions to show the number of bits needed to represent the elements of this space with a desired accuracy. For $f \in \{r, c, s\}$, let $\mathcal{F} = \{f_D, D \in [0, 1]^{n \times d}\}$, be the set of all possible query functions for $d$-dimensional datasets of size $n$. Then, for a norm $\|.\|$, $\mathcal{M} = (\mathcal{F}, \|.\|)$ is a metric space. A learned model, $\hat{f}(.; \theta)$, represents the elements of $\mathcal{M}$ with its parameters $\theta$. Therefore, $\theta$ needs to be large enough (contain enough number of bits), to able to represent *all* the possible elements of $\mathcal{F}$ with a desired accuracy. This in turn depends how *large* $\mathcal{M}$ is and how far its elements are from each other. Thus, our bounds follow from bounding suitable measures of the size of $\mathcal{M}$, specifically, the packing entropy and metric entropy of $\mathcal{M}$. Bounding the packing entropy and metric entropy of $\mathcal{M}$ is non-trivial, especially when considering the 1-norm error, and requires an in-depth analysis of the metric space, relating combinatorial and approximation theoretic concepts. We note that the packing entropy argument is, in essence, the same argument also used by Wei & Yi (2018) in obtaining (part of) their lower bounds (see Sec. 5 for the difference in our results).

Intutiveily, our proofs for the lower bounds on the required model size proceed as follows. Note that if a model has size $\sigma$ bits there are exactly $2^\sigma$ different possible model parameter settings. Thus, when there are more than $2^\sigma$ different possible datasets, multiple datasets must be mapped to the same model parameters values (that is, after training, multiple datasets will have the exact same model parameter values). To obtain a bound on the error, our proof shows that some datasets that are mapped to the same parameter values will be too different from each other (i.e., queries on the datasets have answers that are too different), so that the exact same parameter setting cannot lead to answering queries accurately for both datasets. The proofs do this by constructing $2^\sigma + 1$ different datasets such that query answers differ by more than $2\epsilon$ on all the $2^\sigma + 1$ datasets. Thus, two of those datasets must be mapped to the same model parameter values, and the model must have an error more than $\epsilon$ on at least one of them, which completes the proof. The majority of the proofs are on how this set of $2^\sigma + 1$ datasets is constructed. Specifically, the proofs construct a set of datasets, where each pair differs in some set of points, $S$. The main technical challenge is constructing/showing that for any of the two datasets that differ in a set of points $S$, the maximum or average difference between the query answers is at least $2\epsilon$. This last statement is query dependent, and our technical Lemmas 3-8 in the appendix are proven to quantify the difference between query answers between the datasets based on the properties of the set $S$ and the query type. This is especially difficult in the average-case scenario, as one needs to study how the set $S$ affects all possible queries.

### B.2  BACKGROUND

Our proofs are based on the notions of metric and packing entropy which we briefly describe here.

Consider a metric space, $\mathcal{M} = (\mathcal{F}, \|.\|)$. Let $\mathbb{M}(\mathcal{F}, \|.\|, \epsilon)$ be the packing number of $\mathcal{M}$ and let $\mathbb{N}(\mathcal{F}, \|.\|, \epsilon)$ be the covering number of $\mathcal{M}$. The packing number is the maximum number of non-overlapping balls of radius $\epsilon$ that can be placed in $\mathcal{M}$, and the covering number is the minimum number of balls of radius $\epsilon$ that can cover $\mathcal{M}$ (see Vershynin (2018) for rigorous definitions). We have that for $M = \mathbb{M}(\mathcal{F}, \|.\|, \epsilon)$, that there exists $f_1, ..., f_M \in \mathcal{F}$ s.t. $\|f_i, -f_j\| > \epsilon$ for all $i \neq j$, and for $N = \mathbb{N}(\mathcal{F}, \|.\|, \epsilon)$, that there exists $f_1, ..., f_N \in \mathcal{F}$ s.t. $\forall f \in \mathcal{F}, \|f - f_i\| \leq \epsilon$ for some $i \in [N]$. $\log_2 \mathbb{M}(\mathcal{F}, \|.\|, \epsilon)$ is called the packing entropy of $\mathcal{M}$ and $\log_2 \mathbb{N}(\mathcal{F}, \|.\|, \epsilon)$ is called the metric entropy of $\mathcal{M}$.

Our proofs are based on the following theorem, utilizing the metric and packing entropy of a metric space. Let $\rho_\sigma(f) : \mathcal{F} \to \{0, 1\}^\sigma$ be a function that takes elements of the metric space as input and outputs a bit vector of size $\sigma$, and let $\hat{f}_\sigma : \{0, 1\}^\sigma \to \mathcal{F}'$, be a function that takes bit vector of size $\sigma$ as input and outputs elements of space $\mathcal{F}'$ where $\|.\|$ is well-defined on $\mathcal{F}' \cup \mathcal{F}$.

**Theorem 6.** *Consider a metric space, $\mathcal{M} = (\mathcal{F}, \|.\|)$*

*(i) For any $\rho_\sigma$ and $\hat{f}_\sigma$ such that $\|\hat{f}_\sigma(\rho_\sigma(f)) - f\| \leq \epsilon$ for all $f \in \mathcal{F}$, we must have $\sigma \geq \log_2 \mathbb{M}(\mathcal{F}, \|.\|, 2\epsilon)$.*

*(ii) There exists $\rho_\sigma$ and $\hat{f}_\sigma$ such that $\|\hat{f}_\sigma(\rho_\sigma(f)) - f\| \leq \epsilon$ for all $f \in \mathcal{F}$ with $\sigma \leq \lceil \log_2 \mathbb{N}(\mathcal{F}, \|.\|, \epsilon) \rceil$.*

*Proof of Part (i).* Let $M = \mathbb{M}(\mathcal{F}, \|.\|, 2\epsilon)$. By definition, there exists $f_1, ..., f_M \in \mathcal{F}$, such that $\|f_i - f_j\| > 2\epsilon$ for all $i \neq j$. Now assume, for the purpose of contradiction that there exists $\rho_\sigma$ and $\hat{f}_\sigma$ with $2^\sigma < M$ s.t. $\|\hat{f}_\sigma(\rho_\sigma(f)) - f\| \leq \epsilon$ for all $f \in \mathcal{F}$. Note that a bit vector of size $\sigma$ can take at most $2^\sigma$ different values. Since $2^\sigma < M$, then $\rho(f)$ must create an identical output for at least two of $f_1, ..., f_M$. That is, there exist $f_i$ and $f_j$, $i, j \in [M]$, $i \neq j$ s.t. $\rho(f_i) = \rho(f_j)$. Therefore, $\hat{f}(\rho(f_i)) = \hat{f}(\rho(f_j))$. By assumption the error of approximation is less than $\epsilon$ for both $f_i$ and $f_j$, i.e., $\|\hat{f}(\rho(f_i)) - f_i\| \leq \epsilon$ and $\|\hat{f}(\rho(f_j)) - f_j\| \leq \epsilon$. Then

$$\|f_j - f_i\| \leq \|f_i - \hat{f}(\rho(f_i))\| + \|\hat{f}(\rho(f_i)) - f_j\| = \|f_i - \hat{f}(\rho(f_i))\| + \|\hat{f}(\rho(f_j)) - f_j\| \leq 2\epsilon,$$

Showing $\|f_j - f_i\| \leq 2\epsilon$ which is a contradiction. Therefore, we must have $2^\sigma \geq M$ which implies $\sigma \geq \log_2 \mathbb{M}(\mathcal{F}, \|.\|, 2\epsilon)$ as desired. $\square$

*Proof of Part (ii).* Let $N = \mathbb{N}(\mathcal{F}, \|.\|, \epsilon)$. There exists $f_1, ..., f_N \in \mathcal{F}$ s.t. $\forall f \in \mathcal{F}, \|f - f_i\| \leq \epsilon$ for some $i \in [N]$. Then, construct $\rho_\sigma$ as follows. For any $f \in \mathcal{F}$, find $i$ s.t., $\|f - f_i\| \leq \epsilon$. Let $\rho_\sigma$ be the binary representation of $i$. Since $i \in [N]$, $\lceil \log_2 N \rceil$ bits are needed to represent $i$, so that $\sigma = \lceil \log_2 N \rceil$. Then, for a binary representation $b$, define $\hat{f}_\sigma(b)$ as a function that finds the integer $i$ with representation $b$ and returns $f_i$. Thus, for any $f \in \mathcal{F}$, we have that $\|\hat{f}_\sigma(\rho_\sigma(f)) - f\| = \|f_i - f\| \leq \epsilon$, which completes the proof. $\square$

## B.3 RESULTS WITH $\infty$-NORM

### B.3.1 PROOF OF THEOREM 1

For the purpose of this section, define $\bar{\epsilon} = \lfloor \epsilon \rfloor + 1$ for the proof of all three parts.

*Proof of Part (i).* Let $\mathcal{F} = \{r_D, D \in \mathcal{D}_u^n\}$, be the set of all possible rank functions for datasets of size $n$, and consider the metric space $\mathcal{M} = (\mathcal{F}, \|.\|_\infty)$. We prove a lower bound on $\mathbb{M}(\mathcal{F}, \|.\|_\infty, \epsilon)$ which in turn proves the desired results using Theorem 6.

Let $P_1, P_2 \subseteq \mathcal{D}_u^{\lfloor \frac{n}{\bar{\epsilon}} \rfloor}$, $P_1 \neq P_2$, that is, $P_1$ and $P_2$ are datasets of size $\lfloor \frac{n}{\bar{\epsilon}} \rfloor$ only containing points in $\mathcal{D}_u$. Let $D_1$ be the dataset of size $n$, where each point in $P_1$ is repeated $\bar{\epsilon}$ times (let the remainder of $n - \lfloor \frac{n}{\bar{\epsilon}} \rfloor \times \bar{\epsilon}$ elements be equal to one), and similarly define $D_2$. We have that $\|r_{D_1} - r_{D_2}\|_\infty \geq \bar{\epsilon} > \epsilon$. Let $S$ be the set of all possible datasets generated using the above procedure. We have that $S$ is an $\epsilon$-Packing of $\mathcal{M}$, and thus, $|S| \leq \mathbb{M}(\mathcal{F}, \|.\|_\infty, \epsilon)$.

It remains to find $|S|$. Each element of $S$ is created by selecting $\lfloor \frac{n}{\bar{\epsilon}} \rfloor$ elements from $u + 1$ elements in $\mathcal{D}_u$ with repetition. The unique number of ways to perform this selection is $C(\lfloor \frac{n}{\bar{\epsilon}} \rfloor + u, \lfloor \frac{n}{\bar{\epsilon}} \rfloor) \geq (\frac{\frac{n}{\bar{\epsilon}} + u}{\frac{n}{\bar{\epsilon}}})^{\frac{n}{\bar{\epsilon}}} \geq (\frac{\frac{n}{\epsilon+1} + u}{\frac{n}{\epsilon+1}})^{\frac{n}{\epsilon+1}}$. Thus, $\mathbb{M}(\mathcal{F}, \|.\|_\infty, \epsilon) \geq (1 + \frac{(\epsilon+1)u}{n})^{\frac{n}{\epsilon+1}}$

Combining this with Theorem 6, we have that any model answering queries to $\infty$-error $\epsilon$ must have size at least $\frac{n}{2\epsilon+1} \log_2(1 + \frac{(2\epsilon+1)u}{n})$. $\square$

*Proof of Part (ii).* Let $\mathcal{F} = \{c_D, D \in \mathcal{D}_u^n\}$, be the set of all possible cardinality functions for datasets of size $n$, and consider the metric space $\mathcal{M} = (\mathcal{F}, \|.\|_\infty)$. We prove a lower bound on $\mathbb{M}(\mathcal{F}, \|.\|_\infty, \epsilon)$ which in turn proves the desired results using Theorem 6.

Let $P_1, P_2 \subseteq \mathcal{D}_u^{\lfloor \frac{n}{\bar{\epsilon}} \rfloor \times d}$, $P_1 \neq P_2$, that is, $P_1$ and $P_2$ are $d$-dimensional datasets of size $\lfloor \frac{n}{\bar{\epsilon}} \rfloor$ only containing values in $\mathcal{D}_u$. Let $D_1$ be the dataset of size $n$, where each point in $P_1$ is repeated $\bar{\epsilon}$ times (let the remainder of $n - \lfloor \frac{n}{\bar{\epsilon}} \rfloor \times \bar{\epsilon}$ elements be equal to one), and similarly define $D_2$.

First, we show that $\|c_{D_1} - c_{D_2}\|_\infty \geq \bar{\epsilon}$. Let $p$ be a point that appears more times in $D_1$ than $D_2$, and consider a query $q = (c, r)$ with $c = p - \frac{1}{2u}$ and $r = \frac{1}{u}$. The only point in $P_1 \cup P_2$ that matches $q$ is $p$. Since $p$ appears more times in $D_1$ than $D_2$, and each appearance of the point is repeated $\bar{\epsilon}$ times by definition we have $|c_{D_1}(q) - c_{D_2}(q)| \geq \bar{\epsilon}$.

Thus, we have $\|c_{D_1} - c_{D_2}\|_\infty \geq \bar{\epsilon} > \epsilon$. Let $S$ be the set of all possible datasets generated using the above procedure. We have that $S$ is an $\epsilon$-Packing of $\mathcal{M}$, and thus, $|S| \leq \mathbb{M}(\mathcal{F}, \|.\|_\infty, \epsilon)$.

It remains to find $|S|$. Each element of $S$ is created by selecting $\lfloor \frac{n}{\epsilon} \rfloor$ elements from $(u+1)^d$ elements in $\mathcal{D}_u$ with repetition. The unique number of ways to perform this selection is at least $C(\lfloor \frac{n}{\epsilon} \rfloor + u^d, \lfloor \frac{n}{\epsilon} \rfloor) \geq (\frac{\frac{n}{\epsilon} + u^d}{\frac{n}{\epsilon}})^{\frac{n}{\epsilon}} \geq (\frac{\frac{n}{\epsilon+1} + u^d}{\frac{n}{\epsilon+1}})^{\frac{n}{\epsilon+1}}$. Thus, $\mathbb{IM}(\mathcal{F}, \|.\|_\infty, \epsilon) \geq (1 + \frac{(\epsilon+1)u^d}{n})^{\frac{n}{\epsilon+1}}$.

Combining this with Theorem 6, we have that any model answering queries to $\infty$-error $\epsilon$ must have size at least $\frac{n}{2\epsilon+1} \log_2(1 + \frac{(2\epsilon+1)u^d}{n})$. $\qquad\qquad\square$

*Proof of Part (iii).* For the purpose of contradiction, assume there exists a training/inference function pair $(\rho, \hat{f})$ with size less than $\frac{n}{2\epsilon+1} \log_2(1 + \frac{(2\epsilon+1)u^d}{n})$ that for all datasets $\boldsymbol{D} \in \mathcal{D}_u^{n \times (d+1)}$ we have $\|\hat{f}(.; \rho(\boldsymbol{D})) - s_D\|_\infty \leq \epsilon$. We use $(\rho, \hat{f})$ to construct a training/inference function pair $(\rho', \hat{f})$ that answers cardinality estimation queries for any dataset $\boldsymbol{D} \in \mathcal{D}_u^{n \times d}$ with error at most $\epsilon$. Specifically, define $\rho'(\boldsymbol{D})$ as a function that takes $\boldsymbol{D} \in \mathcal{D}_u^{n \times d}$ as an input, constructs $\boldsymbol{D}' \in \mathcal{D}_u^{n \times (d+1)}$ as $\boldsymbol{D}'[i,j] = \boldsymbol{D}[i,j]$ for $j \in [d], i \in [n]$, and $\boldsymbol{D}'[i, d+1] = 1$, and returns $\rho(\boldsymbol{D}')$. Here, $\boldsymbol{D}'$ is a dataset with its first $d$ dimensions identical to $\boldsymbol{D}$ and but with it's $d+1$-th dimension set to 1 for all data points. Then, we have that $\|\hat{f}(.; \rho'(\boldsymbol{D})) - c_D\|_\infty \leq \epsilon$ for all $\boldsymbol{D} \in \mathcal{D}_u^{n \times d}$, because by construction $c_D = s_{D'}$, $\hat{f}(.; \rho'(\boldsymbol{D})) = \hat{f}(.; \rho(\boldsymbol{D}'))$ and that $\|\hat{f}(.; \rho(\boldsymbol{D}')) - s_{D'}\|_\infty \leq \epsilon$ by assumption. However, $\|\hat{f}(.; \rho'(\boldsymbol{D})) - c_D\|_\infty \leq \epsilon$ contradicts Part (ii), and thus we have proven that no training/inference function pair $(\rho, \hat{f})$ with size less than $\frac{n}{2\epsilon+1} \log_2(1 + \frac{(2\epsilon+1)u^d}{n})$ exists that for all datasets $\boldsymbol{D} \in \mathcal{D}_u^{n \times (d+1)}$ we have $\|\hat{f}(.; \rho(\boldsymbol{D})) - s_D\|_\infty \leq \epsilon$. $\qquad\square$

### B.3.2 Proof of Lemma 1

For any $\rho_\sigma, \hat{f}_\sigma$ with finite $\sigma$, we construct a dataset, $\boldsymbol{D}$, such that $\|r_D - \hat{f}_\sigma(\rho_\sigma(\boldsymbol{D}))\|_\infty > \frac{n}{2}$

Let $\boldsymbol{D}^p$ be the dataset of size $n$ with point $p$ repeated $n$ times. For any $k$, consider $\Delta_k = \{\boldsymbol{D}^{\frac{i}{k}}, 0 \leq i \leq k\}$. Note that for any $\boldsymbol{D}, \boldsymbol{D}' \in \Delta_k$, $\|r_D - r_{D'}\|_\infty = n$. Now for the purpose of contradiction, assume, $\sigma$ bits are sufficient for answering queries with error $\epsilon$ across all datasets of size $n$. Now consider any $\rho_\sigma, \hat{f}_\sigma$. Let $k = 2^\sigma$, so that $|\Delta_k| = 2^\sigma + 1$. Thus, there exists $\boldsymbol{D}, \boldsymbol{D}' \in \Delta_k$ s.t. $\rho_\sigma(\boldsymbol{D}) = \rho_\sigma(\boldsymbol{D}')$, so that $\hat{f}_\sigma(\rho_\sigma(\boldsymbol{D})) = \hat{f}_\sigma(\rho_\sigma(\boldsymbol{D}'))$. Now assume the error on either $\boldsymbol{D}$ or $\boldsymbol{D}'$ is less than $\frac{n}{2}$, or otherwise the proof is complete. Therefore, w.l.o.g., we have $\|\hat{f}_\sigma(\rho_\sigma(\boldsymbol{D})) - r_D\|\infty < \frac{n}{2}$. We have that

$$n = \|r_D - r_{D'}\|_\infty = \|(r_D - \hat{f}_\sigma(\rho_\sigma(\boldsymbol{D}))) - (r_{D'} - \hat{f}_\sigma(\rho_\sigma(\boldsymbol{D}')))\|_\infty < \frac{n}{2} + \|r_{D'} - \hat{f}_\sigma(\rho_\sigma(\boldsymbol{D}'))\|_\infty$$

So that $\|r_{D'} - \hat{f}_\sigma(\rho_\sigma(\boldsymbol{D}'))\|_\infty > \frac{n}{2}$. Thus, for any $\rho_\sigma, \hat{f}_\sigma$ with finite $\sigma$, there exists a dataset, $\boldsymbol{D}$ such that $\|r_D - \hat{f}_\sigma(\rho_\sigma(\boldsymbol{D}))\|_\infty > \frac{n}{2}$. $\qquad\square$

### B.4 Results with 1-norm

### B.4.1 Proof of Theorem 2

We first present the following lemma, whose proof can be found in Appendix C.

**Lemma 3.** *Let $\boldsymbol{D}, \boldsymbol{D}' \in [0,1]^n$ be datasets in sorted order. Then, $\|r_D - r_{D'}\|_1 = \sum_{i \in [n]} |\boldsymbol{D}[i] - \boldsymbol{D}'[i]|$. Note that $\sum_{i \in [n]} |\boldsymbol{D}[i] - \boldsymbol{D}'[i]| = \|\boldsymbol{D} - \boldsymbol{D}'\|_1$, so that $\|r_D - r_{D'}\|_1 = \|\boldsymbol{D} - \boldsymbol{D}'\|_1$.*

The lemma shows that 1-norm between rank functions has a closed-form solution that can be calculated based on the difference between the points in the dataset. We use this lemma throughout for analyzing the 1-norm error.

Let $\mathcal{F} = \{r_D, \boldsymbol{D} \in [0,1]^n\}$, be the set of all possible rank functions for datasets of size $n$, and consider the metric space $\mathcal{M} = (\mathcal{F}, \|.\|_1)$.

*Proof of Theorem 2 (i).* We prove a lower bound on $\mathbb{IM}(\mathcal{F}, \|.\|_1, \epsilon)$ which in turn proves the desired results using Theorem 6.

Let $P = \{\frac{i}{\lceil \frac{k}{\epsilon} \rceil - 1}, 0 \leq i \leq \lceil \frac{k}{\epsilon} \rceil - 1\}$ for an integer $k$ specified later. Let $\boldsymbol{P}, \boldsymbol{P}' \in P^{\lfloor \frac{n}{k} \rfloor}$, $\boldsymbol{P} \neq \boldsymbol{P}'$, that is, $\boldsymbol{P}$ and $\boldsymbol{P}'$ are datasets of size $\lfloor \frac{n}{k} \rfloor$ only containing points in $P$. Let $\boldsymbol{D}$ be the dataset of size

$n$, where each point in $P$ is repeated $k$ times (let the remainder of $n - \lfloor \frac{n}{k} \rfloor \times k$ elements be equal to one), and similarly define $D'$ using $P'$, and consider $D$ and $D'$ in a sorted order.

We show that $\|r_D - r_{D'}\|_1 > \epsilon$. Observe that $D$ and $D'$ differ in at least $k$ points, so that $D[i] \neq D'[i]$ for $k$ different values of $i$. Using this observation and Lemma 3, we have that

$$\|r_D - r_{D'}\|_1 = \|D - D'\|_1 \geq \frac{1}{\lceil \frac{k}{\epsilon} \rceil - 1} \times k > \frac{k}{\frac{k}{\epsilon}} = \epsilon.$$

Thus, for any two different datasets, $D, D'$ generated by the above procedure, we have $\|r_D - r_{D'}\| > \epsilon$. Let $S$ be the set of all datasets generated that way. We have that $S$ is an $\epsilon$-Packing of $\mathcal{M}$, and thus, $|S| \leq \mathbb{M}(\mathcal{F}, \|.\|_1, \epsilon)$.

It remains to find $|S|$. Each element of $S$ is created by selecting $\lfloor \frac{n}{k} \rfloor$ elements from $\lceil \frac{k}{\epsilon} \rceil$ elements in $P$ with repetition. The unique number of ways to perform this selection is $C(\lfloor \frac{n}{k} \rfloor + \lceil \frac{k}{\epsilon} \rceil - 1, \lfloor \frac{n}{k} \rfloor)$. Let $k = \lceil \sqrt{n} \rceil$, so that we have

$$\begin{aligned}
C(\lfloor \frac{n}{\lceil \sqrt{n} \rceil} \rfloor + \lceil \frac{\lceil \sqrt{n} \rceil}{\epsilon} \rceil - 1, \lfloor \frac{n}{\lceil \sqrt{n} \rceil} \rfloor) &\geq (\frac{\lfloor \frac{n}{\lceil \sqrt{n} \rceil} \rfloor + \lceil \frac{\lceil \sqrt{n} \rceil}{\epsilon} \rceil - 1}{\lfloor \frac{n}{\lceil \sqrt{n} \rceil} \rfloor})^{\lfloor \frac{n}{\lceil \sqrt{n} \rceil} \rfloor} \\
&\geq (\frac{\sqrt{n} + \frac{\sqrt{n}}{\epsilon} - 1}{\sqrt{n}})^{\lfloor \frac{n}{\lceil \sqrt{n} \rceil} \rfloor} = (1 + \frac{1}{\epsilon} - \frac{1}{\sqrt{n}})^{\lfloor \frac{n}{\lceil \sqrt{n} \rceil} \rfloor} \\
&\geq (1 + \frac{1}{\epsilon} - \frac{1}{\sqrt{n}})^{\sqrt{n} - 2}.
\end{aligned}$$

Thus, $\mathbb{M}(\mathcal{F}, \|.\|_1, \epsilon) \geq (1 + \frac{1}{\epsilon} - \frac{1}{\sqrt{n}})^{\sqrt{n} - 2}$. Combining this with Theorem 6, we have that any model answering queries to 1-error $\epsilon$ must have size at least $(\sqrt{n} - 2) \log_2(1 + \frac{1}{2\epsilon} - \frac{1}{\sqrt{n}})$. $\qquad\square$

*Proof of Theorem 2 (ii).* We prove an upper bound on the metric entropy $\mathbb{N}(\mathcal{F}, \|.\|_1, \epsilon)$.

Let $\bar{\mathcal{D}} = \{\frac{i}{\lceil \frac{n}{\epsilon} \rceil}, 0 \leq i \leq \lceil \frac{n}{\epsilon} \rceil\}$, and define $\bar{\mathcal{F}} = \{r_D, D \in \bar{\mathcal{D}}^n\}$. We show that $\bar{\mathcal{F}}$ is an $\epsilon$-cover of $\mathcal{F}$, so that its size provides an upper bound for the covering number of $\mathcal{F}$.

Specifically, for any $D \in [0, 1]^n$, we show that $\exists r_{\bar{D}} \in \bar{\mathcal{F}}$, s.t. $\|r_D - r_{\bar{D}}\|_1 \leq \epsilon$. Consider $\bar{D}$ such that $\bar{D}[i] = \frac{\lfloor \lceil \frac{n}{\epsilon} \rceil \times D[i] \rfloor}{\lceil \frac{n}{\epsilon} \rceil}$ for all $i$. Such a $\bar{D}$ exists in $\bar{\mathcal{D}}^n$ as all its points belong to $\bar{\mathcal{D}}$. This is because $\lfloor \lceil \frac{n}{\epsilon} \rceil \times D[i] \rfloor$ is an integer between 0 and $\lceil \frac{n}{\epsilon} \rceil$, for $D[i] \in [0, 1]$. Furthermore, using Lemma 3,

$$\begin{aligned}
\|r_D - r_{\bar{D}}\|_1 &= \sum_{i \in [n]} |D[i] - \bar{D}[i]| \\
&= \sum_{i \in [n]} |D[i] - \frac{\lfloor \lceil \frac{n}{\epsilon} \rceil \times D[i] \rfloor}{\lceil \frac{n}{\epsilon} \rceil}| \\
&< \sum_{i \in [n]} \frac{1}{\lceil \frac{n}{\epsilon} \rceil} \\
&= \frac{n}{\lceil \frac{n}{\epsilon} \rceil} \leq \epsilon.
\end{aligned}$$

Therefore, $\bar{\mathcal{F}}$ is an $\epsilon$-covering of $\mathcal{F}$. It remains to calculate the size of $\bar{\mathcal{F}}$, which is the number of possible ways to select $n$ elements from $\lceil \frac{n}{\epsilon} \rceil + 1$ elements in $\bar{\mathcal{D}}$ with repetition. That is,

$$|\bar{\mathcal{F}}| = C(n + \lceil \frac{n}{\epsilon} \rceil, n) \leq (e\frac{n + \lceil \frac{n}{\epsilon} \rceil}{n})^n \leq (e\frac{n + \frac{n}{\epsilon} + 1}{n})^n = (e + \frac{e}{\epsilon} + \frac{e}{n})^n,$$

So that $\mathbb{N}(\mathcal{F}, \|.\|_1, \epsilon) \leq (e + \frac{e}{\epsilon} + \frac{e}{n})^n$. Combining this with Theorem 6, we obtain that $n \log_2(e + \frac{e}{\epsilon} + \frac{e}{n})$ is an upper bound on the required model size. $\qquad\square$

B.4.2 PROOF OF THEOREM 3

We first present two lemmas, whose proof can be found in Appendix C. These lemmas can be seen as substitute for Lemma 3 for the case of cardinality estimation, since we do not have such a closed-form general statement for 1-norm difference between cardinality functions (as we did for indexing in Lemma 3). However, the following lemmas provide scenarios where the 1-norm difference between the cardinality functions are bounded, which are utilized in our Theorem's proof.

**Lemma 4.** *Consider two 1-dimensional databases $\boldsymbol{D}$ and $\boldsymbol{D}'$ of size $n$ such that $|\boldsymbol{D}[i] - \boldsymbol{D}'[i]| \leq \frac{\epsilon}{n}$ for $i \in [n]$. Then $\|c_D - c_{D'}\|_1 \leq 2\epsilon$.*

**Lemma 5.** *Consider a 1-dimensional database $\boldsymbol{D}$ of size $n$. Assume that we are given two mask vectors, $\boldsymbol{m}^1, \boldsymbol{m}^2 \in \{0, 1\}^n$ that each create two new dataset $\boldsymbol{D}^1$ and $\boldsymbol{D}^2$, s.t., $\boldsymbol{D}^i$ consists of records in $\boldsymbol{D}$ with $\boldsymbol{m}^i = 1$ for $i \in \{1, 2\}$. We have that $\|c_{D^1} - c_{D^2}\|_1 \leq \frac{1}{2} \sum_{i \in [n]} |\boldsymbol{m}^1[i] - \boldsymbol{m}^2[i]|$*

For the remainder of this section, let $\mathcal{F} = \{c_D, \boldsymbol{D} \in [0, 1]^{n \times d}\}$, be the set of all possible cardinality functions for $d$-dimensional datasets of size $n$, and consider the metric space $\mathcal{M} = (\mathcal{F}, \|.\|_1)$. Our proof in this case reduce the problem to a 1-dimensional setting and then utilize the lemmas stated above to analyze the cardinality functions.

*Proof of Theorem 3 Part (i).* We prove a lower bound on $\mathbb{M}(\mathcal{F}, \|.\|_1, \epsilon)$ which in turn proves the desired results using Theorem 6.

Let $\frac{u}{2} = \lceil \frac{k}{2\epsilon} \rceil - 1$ and let $P = \{(\frac{i_1}{u}, ..., \frac{i_d}{u}), 0 \leq i_1, ..., i_d \leq \frac{u}{2}\}$ for an integer $k$ specified later. Let $\boldsymbol{P}, \boldsymbol{P}' \in P^{\lfloor \frac{n}{k} \rfloor}$, $\boldsymbol{P} \neq \boldsymbol{P}'$, that is, $\boldsymbol{P}$ and $\boldsymbol{P}'$ are datasets of size $\lfloor \frac{n}{k} \rfloor$ only containing points in $P$. Let $\boldsymbol{D}$ be the dataset of size $n$, where each point in $\boldsymbol{P}$ is repeated $k$ times, and similarly define $\boldsymbol{D}'$ using $\boldsymbol{P}'$. W.l.o.g, assume $\boldsymbol{D}$ and $\boldsymbol{D}'$ differ on their $i$-th point and its $d$-th dimension. We have

$$\|c_D - c_{D'}\|_1 = \int_{c_1} ... \int_{c_{d-1}} \int_{r_1} ... \int_{r_{d-1}} \|c_D(c_1, ..., c_{d-1}, ., r_1, ..., r_{d-1}, .) - c_{D'}(c_1, ..., c_{d-1}, ., r_1, ..., r_{d-1}, .)\|_1$$

$$\geq \int_{c_1} ... \int_{c_{d-1}} \int_{r_1} ... \int_{r_{d-1}} \mathcal{I}_{\boldsymbol{q}, \boldsymbol{D}[i]}^{d-1} \|c_D(c_1, ..., c_{d-1}, ., r_1, ..., r_{d-1}, .) - c_{D'}(c_1, ..., c_{d-1}, ., r_1, ..., r_{d-1}, .)\|_1$$

Where $\mathcal{I}_{\boldsymbol{q}, \boldsymbol{p}}^i$ is an indicator function equal to 1 if a record $\boldsymbol{p}$ matches the first $i$ dimensions in $\boldsymbol{q} = (c_1, ..., c_d, r_1, ..., r_d)$, that is if $c_j \leq p_j \leq c_j + r_j$ for all $j \in [i]$, and zero otherwise.

Next, we show that for any $\boldsymbol{q}$, we have

$$\mathcal{I}_{\boldsymbol{q}, \boldsymbol{D}[i]}^{d-1} \|c_D(c_1, ..., c_{d-1}, ., r_1, ..., r_{d-1}, .) - c_{D'}(c_1, ..., c_{d-1}, ., r_1, ..., r_{d-1}, .)\|_1 \geq \mathcal{I}_{\boldsymbol{q}, \boldsymbol{D}[i]}^{d-1} \frac{\epsilon}{2}.$$

Specifically, if $\boldsymbol{D}[i]$ does not match the first $d - 1$ dimensions of $\boldsymbol{q}$, then both sides are zero. Otherwise, let $\bar{\boldsymbol{D}} = \{\boldsymbol{D}[i, d] \ \forall i, \mathcal{I}_{\boldsymbol{q}, \boldsymbol{D}[i]}^{d-1} = 1\}$ be a 1-dimensional dataset of the $d$-th attribute of the records of $\boldsymbol{D}$ that matches the first $d - 1$ dimensions of $\boldsymbol{q}$, and similarly define $\bar{\boldsymbol{D}}' = \{\boldsymbol{D}'[i, d] \ \forall i, \mathcal{I}_{\boldsymbol{q}, \boldsymbol{D}'[i]}^{d-1} = 1\}$ for $\boldsymbol{D}'$. By, definition, we have that

$$\|c_D(c_1, ..., c_{d-1}, ., r_1, ..., r_{d-1}, .) - c_{D'}(c_1, ..., c_{d-1}, ., r_1, ..., r_{d-1}, .)\|_1 = \|c_{\bar{D}} - c_{\bar{D}'}\|_1.$$

We state the following lemma whose proof is deferred to Appendix C to bound the 1-norm difference between $\bar{\boldsymbol{D}}$ and $\bar{\boldsymbol{D}}'$.

**Lemma 6.** *Let $\bar{\boldsymbol{D}}$ and $\bar{\boldsymbol{D}}'$ be as defined above. For $\epsilon < k$, we have that $\|c_{\bar{D}} - c_{\bar{D}'}\|_1 > \frac{\epsilon}{2}$.*

Using Lemma 6 and the preceding discussion, we have that

$$\|c_D - c_{D'}\|_1 > \int_{c_1} ... \int_{c_{d-1}} \int_{r_1} ... \int_{r_{d-1}} \mathcal{I}_{\boldsymbol{q}, \boldsymbol{D}[i]}^{d-1} \frac{\epsilon}{2}$$

$$= \frac{\epsilon}{2} \int_{c_1} ... \int_{c_{d-1}} \int_{r_1} ... \int_{r_{d-1}} \mathcal{I}_{\boldsymbol{q}, \boldsymbol{D}[i]}^{d-1}.$$

We have that $\mathcal{I}_{\boldsymbol{q},\boldsymbol{D}[i]}^{d-1} = 1$ for queries where, for all dimension, $j$, $r_j \in [0,1]$ and $c_j \in [\boldsymbol{D}[i,j], \min\{\boldsymbol{D}[i,j], 1 - r_j\}]$. Calculating the total volume of such queries, we have

$$
\int_0^1 \int_{\boldsymbol{D}[i,j]-r_j}^{\min\{\boldsymbol{D}[i,j],1-r_j\}} 1 \, \mathrm{d}c_j \, \mathrm{d}r_j = \int_0^1 (\min\{\boldsymbol{D}[i,j], 1 - r_j\} - \boldsymbol{D}[i,j] + r_j) \, \mathrm{d}r_j
$$

$$
= \int_0^{1-\boldsymbol{D}[i,j]} r_j \, \mathrm{d}r_j + \int_{1-\boldsymbol{D}[i,j]}^1 (1 - \boldsymbol{D}[i,j]) \, \mathrm{d}r_j
$$

$$
= \frac{1}{2}(1 - \boldsymbol{D}[i,j])^2 + (1 - \boldsymbol{D}[i,j])^2
$$

$$
= \frac{1}{2} - \frac{\boldsymbol{D}[i,j]^2}{2}
$$

$$
\geq \frac{1}{4},
$$

Where the last inequality follows because all points in $P$ have all of their coordinates less than $\frac{1}{2}$.

Repeating the above for all the dimensions, we obtain that

$$
\|c_D - c_{D'}\|_1 > \epsilon 0.25^d.
$$

Thus, for any two different datasets, $\boldsymbol{D}, \boldsymbol{D}'$ generated by the above procedure, we have $\|c_D - c_{D'}\| > \epsilon 0.25^d$. Let $S$ be the set of all datasets generated that way.

It remains to find $|S|$. Each element of $S$ is created by selecting $\lfloor \frac{n}{k} \rfloor$ from the $(\frac{u}{2} + 1)^d$ possible elements in $P$ with repetition. Set $k = \lceil \sqrt{n} \rceil + 1$, which is

$$
C((\frac{u}{2} + 1)^d + \lceil \sqrt{n} \rceil, \lceil \sqrt{n} \rceil + 1) \geq \left( \frac{(\frac{u}{2} + 1)^d + \lfloor \frac{n}{\lceil \sqrt{n} \rceil} \rfloor - 1}{\lfloor \frac{n}{\lceil \sqrt{n} \rceil} \rfloor} \right)^{\lfloor \frac{n}{\lceil \sqrt{n} \rceil} \rfloor}
$$

$$
\geq \left( \frac{(\frac{\sqrt{n}}{2\epsilon})^d + \sqrt{n} - 1}{\sqrt{n}} \right)^{\sqrt{n}-2}
$$

$$
= \left( \frac{\sqrt{n}^{d-1}}{(2\epsilon)^d} + 1 - \frac{1}{\sqrt{n}} \right)^{\sqrt{n}-2}.
$$

Thus, we have that $S$ contains elements each of which are at least $\epsilon 0.25^d$ apart for $\epsilon < \sqrt{n}$.

Define $\epsilon' = \epsilon 0.25^d$ so that $\frac{\epsilon'}{0.25^d} = \epsilon$, and repeat tha above procedure for $\epsilon'$, we have that there exists a set of $\left( \frac{\sqrt{n}^{d-1}}{4^{d(d+1)}\epsilon^d} + 1 - \frac{1}{\sqrt{n}} \right)^{\sqrt{n}-2}$ elements where they all differ by $\epsilon'$ for $4^d \epsilon' \leq \sqrt{n}$. Thus, we have an $\epsilon$-Packing of $\mathcal{M}$, and thus, $\left( \frac{\sqrt{n}^{d-1}}{4^{d(d+1)}\epsilon^d} + 1 - \frac{1}{\sqrt{n}} \right)^{\sqrt{n}-2} \leq \mathbb{M}(\mathcal{F}, \|.\|_1, \epsilon)$. Using this together with Theorem 6 proves the results. $\qquad \square$

*Proof of Theorem 3 Part (ii).* We prove an upper bound on $\mathbb{N}(\mathcal{F}, \|.\|_1, \epsilon)$ which in turn proves the desired results using Theorem 6.

Let $\bar{\mathcal{D}} = \{(\frac{i_1}{\lceil \frac{n}{\epsilon} \rceil}, ..., \frac{i_d}{\lceil \frac{n}{\epsilon} \rceil}), 0 \leq i_1, ..., i_d \leq \lceil \frac{n}{\epsilon} \rceil\}$, and define $\bar{\mathcal{F}} = \{c_D, \boldsymbol{D} \in \bar{\mathcal{D}}^n\}$. We show that $\bar{\mathcal{F}}$ is an $\epsilon$-cover of $\mathcal{F}$, so that its size provides an upper bound for the covering number of $\mathcal{F}$.

For any $\boldsymbol{D} \in [0,1]^{n \times d}$, we show that $\exists c_{\bar{D}} \in \bar{\mathcal{F}}$, s.t. $\|c_D - c_{\bar{D}}\|_1 \leq \epsilon$. Specifically, consider $\bar{\boldsymbol{D}}$ such that $\bar{\boldsymbol{D}}[i,j] = \frac{\lfloor \lceil \frac{n}{\epsilon} \rceil \times \boldsymbol{D}[i,j] \rfloor}{\lceil \frac{n}{\epsilon} \rceil}$ for all $i$. Such a $\bar{\boldsymbol{D}}$ exists in $\bar{\mathcal{D}}^n$ as all its points belong to $\bar{\mathcal{D}}$. Our goal is to show that $\|c_D - c_{\bar{D}}\|_1 \leq \epsilon$.

To do so, consider the dataset $\hat{\boldsymbol{D}}$ that $\hat{\boldsymbol{D}}[i,j] = \bar{\boldsymbol{D}}[i,j]$ for $j \in [d-1]$ and $\hat{\boldsymbol{D}}[i,d] = \boldsymbol{D}[i,d]$. That is $\hat{\boldsymbol{D}}$ is a dataset with its first $d-1$ dimensions the same as $\bar{\boldsymbol{D}}$ and its $d$-th dimension the same as $\boldsymbol{D}$. We have that

$$
\|c_D - c_{\bar{D}}\|_1 \leq \|c_D - c_{\hat{D}}\|_1 + \|c_{\hat{D}} - c_{\bar{D}}\|_1
$$

We study the first two terms separately. Consider the second term $\|c_{\hat{D}} - c_{\bar{D}}\|_1$. Observe that the first $d-1$ dimensions of $\bar{\boldsymbol{D}}$ and $\hat{\boldsymbol{D}}$ are identical, so that applying any predicate on the first $d-1$

attributes of the datasets leads to the same filtered data points. Furthermore, the $d$-th dimension of the points differ by at most $\frac{\epsilon}{n}$, so by Lemma 4, we have

$$\|c_{\hat{D}} - c_{\bar{D}}\|_1 \leq 2\epsilon.$$

The following Lemma, proven in Appendix C, studies $\|c_D - c_{\hat{D}}\|_1$.

**Lemma 7.** *Let $D$ and $\hat{D}$ be defined as above. We have $\|c_D - c_{\hat{D}}\|_1 \leq (d-1)\epsilon$.*

Thus, putting everything together, we have that

$$\|c_D - c_{\bar{D}}\|_1 \leq 2\epsilon + (d-1)\epsilon = (d+1)\epsilon.$$

We have shown that for any $D$, there exists $c_{\bar{D}} \in \mathcal{F}$ such that $\|c_D - c_{\bar{D}}\| \leq (d+1)\epsilon$. Next, we calculate the size of $\mathcal{F}$. it is equal to the number of ways $n$ elements can be selected from a set of size $(\lceil \frac{n}{\epsilon} \rceil + 1)^d$. This is equal to

$$C((\lceil \frac{n}{\epsilon} \rceil + 1)^d + n - 1, n) \leq (e\frac{(\lceil \frac{n}{\epsilon} \rceil + 1)^d + n - 1}{n})^n$$

Define $\epsilon' = (d+1)\epsilon$, we have that there exists an $\epsilon'$-cover of $\mathcal{F}$ with $(e\frac{(\frac{2n(d+1)}{\epsilon'})^d + n - 1}{n})^n$ elements for $\epsilon' \leq \frac{2}{3}n(d+1)$ (we have used the fact that $2x \geq \lceil x \rceil + 1$ for $x \geq \frac{3}{2}$), which proves $\mathbb{N}(\mathcal{F}, \|.\|_1, \epsilon) \leq (e\frac{(\frac{2n(d+1)}{\epsilon'})^d + n - 1}{n})^n$. Note that $d \geq 1$ implies that $\epsilon' \leq \frac{2}{3}n(d+1)$ is satisfied for all $\epsilon' \leq n$. Combining this with Theorem 6, we obtain that $n \log_2(e(\frac{2(d+1)}{\epsilon})^d n^{d-1} + e - \frac{e}{n})$ is an upper bound on the required model size. $\square$

### B.4.3  PROOFS FOR RANGE-SUM ESTIMATION

Similar to previous sections, we first present the following lemma that is used for bounding difference between range-sum functions, proved in Appendix C. Lemmas 9 and 10 are a direct generalization of Lemmas 4 and 5 to answering range count queries, with almost identical proofs. However, Lemma 8 is specific to range-sum queries, allowing us to analyze the attribute that is being aggregated by the query.

**Lemma 8.** *Assume $D$ and $D'$ are two $(d+1)$-dimensional datasets with identical first $d$ dimensions, but with $|D[i, d+1] - D'[i, d+1]| \leq \frac{\epsilon}{n}$ for $i \in [n]$. Then, $\|s_D - s_{D'}\| \leq \epsilon$.*

**Lemma 9.** *Consider two 2-dimensional databases $D$ and $D'$ of size $n$ such that $|D[i, 1] - D'[i, 1]| \leq \frac{\epsilon}{n}$ and $D[i, 2] = D'[i, 2]$ for $i \in [n]$. Then $\|s_D - s_{D'}\|_1 \leq 2\epsilon$.*

**Lemma 10.** *Consider a 2-dimensional database $D$ of size $n$. Assume that we are given two mask vectors, $m^1, m^2 \in \{0, 1\}^n$ that each create two new dataset $D^1$ and $D^2$, s.t., $D^i$ consists of records in $D$ with $m^i = 1$ for $i \in \{1, 2\}$. We have that $\|s_{D^1} - s_{D^2}\|_1 \leq \frac{1}{2} \sum_{i \in [n]} |m^1[i] - m^2[i]|$.*

In this section, let $\mathcal{F} = \{s_D, D \in [0, 1]^{n \times d}\}$, be the set of all possible range-sum functions for $d + 1$-dimensional datasets of size $n$, and consider the metric space $\mathcal{M} = (\mathcal{F}, \|.\|_1)$.

*Proof of Corollary 1.* For the purpose of contradiction, assume there exists a training/inference function pair $(\rho, \hat{f})$ with size less than $(\sqrt{n} - 2) \log_2(1 + \frac{\sqrt{n}^{d-1}}{4^{d(d+1)}\epsilon^d} - \frac{1}{\sqrt{n}})$ that for all datasets $D \in [0, 1]^{n \times (d+1)}$ we have $\|\hat{f}(.; \rho(D)) - s_D\|_1 \leq \epsilon$. We use $(\rho, \hat{f})$ to construct a training/inference function pair $(\rho', \hat{f})$ that answers cardinality estimation queries for any dataset $D \in [0, 1]^{n \times d}$ with error at most $\epsilon$. Specifically, define $\rho'(D)$ as a function that takes $D \in [0, 1]^{n \times d}$ as an input, constructs $D' \in [0, 1]^{n \times (d+1)}$ as $D'[i, j] = D[i, j]$ for $j \in [d], i \in [n]$, and $D'[i, d+1] = 1$, and returns $\rho(D')$. Here, $D'$ is a dataset with its first $d$ dimensions identical to $D$ and but with it's $d + 1$-th dimension set to 1 for all data points. Then, we have that $\|\hat{f}(.; \rho'(D)) - c_D\|_1 \leq \epsilon$ for all $D \in [0, 1]^{n \times d}$, because by construction $c_D = s_{D'}$ and that $\hat{f}(.; \rho'(D)) = \hat{f}(.; \rho(D'))$ and that $\|\hat{f}(.; \rho(D')) - s_{D'}\|_1 \leq \epsilon$ by assumption. However, $\|\hat{f}(.; \rho'(D)) - c_D\|_1 \leq \epsilon$ contradicts Theorem 3, and thus we have proven that no training/inference function pair $(\rho, \hat{f})$ with size less than $(\sqrt{n} - 2) \log_2(1 + \frac{\sqrt{n}^{d-1}}{4^{d(d+1)}\epsilon^d} - \frac{1}{\sqrt{n}})$ exists that for all datasets $D \in [0, 1]^{n \times (d+1)}$ we have $\|\hat{f}(.; \rho(D)) - s_D\|_1 \leq \epsilon$. Thus, $\Sigma_s^1 \geq (\sqrt{n} - 2) \log_2(1 + \frac{\sqrt{n}^{d-1}}{4^{d(d+1)}\epsilon^d} - \frac{1}{\sqrt{n}})$. $\square$

*Proof of Theorem 4.* Let $\bar{\mathcal{D}} = \{(\frac{i_1}{\lceil \frac{n}{\epsilon} \rceil}, ..., \frac{i_{d+1}}{\lceil \frac{n}{\epsilon} \rceil}), 0 \le i_1, ..., i_{d+1} \le \lceil \frac{n}{\epsilon} \rceil\}$, and define $\bar{\mathcal{F}} = \{s_D, \boldsymbol{D} \in \bar{\mathcal{D}}^n\}$. We show that $\bar{\mathcal{F}}$ is an $\epsilon$-cover of $\mathcal{F}$, so that its size provides an upper bound for the covering number of $\mathcal{F}$. For any $\boldsymbol{D} \in [0,1]^{n \times d}$, we show that $\exists s_{\bar{D}} \in \bar{\mathcal{F}}$, s.t. $\|s_D - s_{\bar{D}}\|_1 \le \epsilon$. Specifically, consider $\bar{\boldsymbol{D}}$ such that $\bar{\boldsymbol{D}}[i,j] = \frac{\lfloor \lceil \frac{n}{\epsilon} \rceil \times \boldsymbol{D}[i,j] \rfloor}{\lceil \frac{n}{\epsilon} \rceil}$ for all $i$. Such a $\bar{\boldsymbol{D}}$ exists in $\bar{\mathcal{D}}^n$ as all its points belong to $\bar{\mathcal{D}}$. Furthermore, define $\boldsymbol{D}'$ as $\boldsymbol{D}'[i,j] = \boldsymbol{D}_{i,j}$ for $i \in [n], j \in [d]$ but $\boldsymbol{D}'[i,d+1] = \bar{\boldsymbol{D}}[i,d+1]$. We have that $\|s_D - s_{\bar{D}}\|_1 \le \|s_D - s_{D'}\|_1 + \|s_{D'} - s_{\bar{D}}\|_1 \le \|s_{D'} - s_{\bar{D}}\|_1 + \epsilon$ by Lemma 8

The remainder of the proof follows closely the proof of Theorem 3 Part (ii) with a slight generalization. We state this as a lemma and differ the proof to Appendix C.

**Lemma 11.** *Let $\bar{\boldsymbol{D}}$ and $\boldsymbol{D}'$ be as defined as above. We have that $\|s_{\bar{\boldsymbol{D}}} - s_{\boldsymbol{D}'}\| \le (d+1)\epsilon$*

This, together with the above discussion implies that $\|s_D - s_{\bar{D}}\| \le (d+2)\epsilon$. Thus, we have shown that for any $\boldsymbol{D}$, there exists $s_{\bar{D}} \in \mathcal{F}$ such that $\|s_D - s_{\bar{D}}\| \le (d+1)\epsilon$. Next, we calculate the size of $\mathcal{F}$. it is equal to the number of ways $n$ elements can be selected from a set of size $(\lceil \frac{n}{\epsilon} \rceil + 1)^{d+1}$. This is equal to

$$C((\lceil \frac{n}{\epsilon} \rceil + 1)^{d+1} + n - 1, n) \le (e\frac{(\lceil \frac{n}{\epsilon} \rceil + 1)^{d+1} + n - 1}{n})^n$$

Define $\epsilon' = (d+2)\epsilon$, we have that there exists an $\epsilon'$-cover of $\mathcal{F}$ with $(e^{\frac{(\frac{2n(d+2)}{\epsilon'})^{d+1} + n - 1}{n}})^n$ elements for $\epsilon' \le \frac{2}{3}n(d+2)$ (we have used the fact that $2x \ge \lceil x \rceil + 1$ for $x \ge \frac{3}{2}$), which proves $\mathbb{N}(\mathcal{F}, \|.\|_1, \epsilon) \le (e^{\frac{(\frac{2n(d+2)}{\epsilon'})^{d+1} + n - 1}{n}})^n$. Note that $d \ge 1$ implies that $\epsilon' \le \frac{2}{3}n(d+2)$ is satisfied for all $\epsilon' \le n$. Combining this with Theorem 6, we obtain that $n \log_2(e(\frac{2(d+2)}{\epsilon})^{d+1}n^d + e - \frac{e}{n})$ is an upper bound on the required model size. $\qquad\square$

## B.5 RESULTS WITH $\mu$-NORM

### B.5.1 PROOF OF THEOREM 5

Theorem 5 can be seen as a direct generalization of Theorem 2. We first present a generalization of Lemma 3 to the case of $\mu$-norm.

**Lemma 12.** *Let $\boldsymbol{D}$ and $\boldsymbol{D}'$ be 1-dimensional datasets in sorted order. Then, $\|r_D - r_{D'}\|_\mu = \|\boldsymbol{D} - \boldsymbol{D}'\|_\mu$, where $\|D_1 - D_2\|_\mu = \sum_{i \in [n]} \mu([D_1[i], D_2[i]])$ and $\mu([D_1[i], D_2[i]])$ is the probability of observing a query in the range $[D_1[i], D_2[i]]$.*

Using this lemma, the remainder of the proof is a straightforward adaptation of arguments in the proof of Theorem 2. Let $\mathcal{F} = \{r_D, \boldsymbol{D} \in [0,1]^n\}$, be the set of all possible rank functions for datasets of size $n$, and consider the metric space $\mathcal{M} = (\mathcal{F}, \|.\|_\mu)$.

*Proof of Theorem 5 Part (i).* Let $p_0 = 0$ and define $p_i$ inductively s.t. $\mu([p_{i-1}, p_i]) = \frac{1}{\lceil \frac{k}{\epsilon} \rceil - 1}$ for $i > 0$. Since $\mu$ is a continuous distribution over $[0,1]$, a total of $\lceil \frac{k}{\epsilon} \rceil$ distinct such points in $[0,1]$ exist. Let $P = \{p_0, ..., p_1\}$ be the set of all such points, for an integer $k$ specified later. Let $\boldsymbol{P}, \boldsymbol{P}' \in P^{\lfloor \frac{n}{k} \rfloor}$, $\boldsymbol{P} \ne \boldsymbol{P}'$, that is, $\boldsymbol{P}$ and $\boldsymbol{P}'$ are datasets of size $\lfloor \frac{n}{k} \rfloor$ only containing points in $P$. Let $\boldsymbol{D}$ be the dataset of size $n$, where each point in $\boldsymbol{P}$ is repeated $k$ times, and similarly define $\boldsymbol{P}'$. Note that $\boldsymbol{D}$ and $\boldsymbol{D}'$ differ in at least $k$ points, so that $\boldsymbol{D}[i] \ne \boldsymbol{D}'[i]$ for $k$ different $i$'s. This means $\|r_D - r_{D'}\|_\mu = \|\boldsymbol{D} - \boldsymbol{D}'\|_\mu \ge \frac{1}{\lceil \frac{k}{\epsilon} \rceil - 1} > \frac{\epsilon}{k}k = \epsilon$. Therefore, as long as we generate datasets the way described above, for every two datasets we have $\|r_D - r_{D'}\| > \epsilon$. Similar to the case of 1-norm error, the total number of such distinct datasets is greater than $(1 + \frac{1}{\epsilon} - \frac{1}{\sqrt{n}})^{\sqrt{n}-2}$ which bounds the packing entropy and together with Theorem 6 proves the results (see proof of Theorem 2 for more details). $\qquad\square$

*Proof of Theorem 5 Part (ii).* Let $p_0 = 0$ and define $p_i$ s.t. $\mu([p_{i-1}, p_i]) = \frac{1}{\lceil \frac{n}{\epsilon} \rceil}$ and let $P$ be the set of $\lceil \frac{n}{\epsilon} \rceil + 1$ such points. For any $\boldsymbol{D} \in [0,1]^n$, we show that $\exists \bar{\boldsymbol{D}} \in P^n$, s.t. $\|\boldsymbol{D} - \bar{\boldsymbol{D}}\|_\mu \le \epsilon$. Specifically, consider $\bar{\boldsymbol{D}}$ such that $\bar{\boldsymbol{D}}[i] = \arg\min_{p \in P} |p - \boldsymbol{D}[i]|$ for all $i$. Note that for every $i$, we have that $\mu([\bar{\boldsymbol{D}}[i], \boldsymbol{D}[i]]) \le \frac{\epsilon}{n}$. Therefore, $\|r_D - r_{\bar{D}}\| \le \sum_{i \in [n]} \frac{\epsilon}{n} = \epsilon$. Similar to the case of 1-norm error, we can calculate the total number of possible datasets in $P^n$ to be at most $(e + \frac{e}{\epsilon} + \frac{e}{n})^n$, which combined with Theorem 6 proves the result. $\qquad\square$

### B.5.2 PROOF OF LEMMA 2

For $f \in \{s, c\}$, we construct a query distribution such that for any error $\epsilon > 0$, we have $\|f_D - f_{D'}\| \leq \epsilon$ for any $D$ and $D'$. Specifically, consider the set $Q = \{(\frac{i}{kn}, \frac{1}{kn}), 0 \leq i \leq kn - 1\}$ for an integer $k$ defined later.

Then, define the p.d.f. as $g(c, r) = 2nk$ if $\forall x \in Q, x \notin [c, c + r]$ and 0 otherwise. Note that this is a valid p.d.f that integrates to 1, as shown below.

$$
\sum_{0 \leq i \leq kn-1} \int_{c=0}^{\frac{1}{nk}} \int_{r=0}^{r=\frac{1}{nk}-c} 2nk \, \mathrm{d}r \, \mathrm{d}c = 2nk \sum_{0 \leq i \leq kn-1} \int_{c=0}^{\frac{1}{nk}} (\frac{1}{nk} - c) \, \mathrm{d}c
$$

$$
= 2nk \sum_{0 \leq i \leq kn-1} [\frac{1}{nk}c - \frac{c^2}{2}]_0^{\frac{1}{nk}}
$$

$$
= 2nk \sum_{0 \leq i \leq kn-1} \frac{1}{nk}^2 - \frac{(\frac{1}{nk})^2}{2}
$$

$$
= \frac{2nk}{2} \sum_{0 \leq i \leq kn-1} (\frac{1}{nk})^2 = 1
$$

Now for any two dataset $D$ and $D'$, we have

$$
\|f_D - f_{D'}\| = 2kn \sum_i \int_{c=\frac{i}{nk}}^{\frac{i+1}{nk}} \int_{r=0}^{r=\frac{i+1}{nk}-c} |f_D(c, r) - f_{D'}(c, r)|
$$

$$
\leq 2kn \sum_i \int_{c=\frac{i}{nk}}^{\frac{i+1}{nk}} \int_{r=0}^{r=\frac{i+1}{nk}-c} 2f_D(c, r)
$$

$$
\leq 4kn \sum_i \int_{c=\frac{i}{nk}}^{\frac{i+1}{nk}} \int_{r=0}^{r=\frac{i+1}{nk}-c} f_D(\frac{i}{nk}, \frac{1}{nk})
$$

$$
= 4kn \sum_i f_D(\frac{i}{nk}, \frac{1}{nk}) \frac{1}{(nk)^2} = \frac{4}{(nk)} \sum_i f_D(\frac{i}{nk}, \frac{1}{nk})
$$

$$
\leq \frac{4}{k}.
$$

The first inequality follows by assuming, w.l.o.g that $f_D(q) > f_{D'}(q)$. The second inequality follows because both cardinality and range-sum functions are increasing functions in the length of the query predicate and the last inequality follows since $\sum_i |f_{D_1}(\frac{i}{nk}, \frac{1}{nk})| \leq n$ because all queries in $Q$ are disjoint, so that every point in $D$ will contribute at most once to queries in $Q$. Now setting $k$ so that $\frac{4}{k} < \epsilon$ complete the proof. $\qquad\square$

## C    Proof of Technical Lemmas

*Proof of Lemma 3.* The 1-norm error is the area between the two curves for $\boldsymbol{D}$ and $\boldsymbol{D}'$. We break down this area into rectangles whose area can be computed in closed form. The main intuition is to calculate this area by considering vertically stacked rectangles on top of each other. Specifically, we have $\|r_D - r_{D'}\|_1 = \int_0^1 |r_D(q) - r_{D'}(q)|$. Let $I_{i,q}$ be an indicator function denoting whether $q \in [\boldsymbol{D}[i], \boldsymbol{D}'[i]]$ (or $[\boldsymbol{D}'[i], \boldsymbol{D}[i]]$ if $\boldsymbol{D}'[i] < \boldsymbol{D}[i]$). We have that $|r_D(q) - r_{D'}(q)| = \sum_{i=0}^n I_{i,q}$. Thus, we have $\|r_D - r_{D'}\|_1 = \int_0^1 \sum_{i=0}^n I_{i,q} = \sum_{i=0}^n \int_0^1 I_{i,q} = \sum_{i=0}^n \int_{\boldsymbol{D}[i]}^{\boldsymbol{D}'[i]} 1 = \sum_i |D[i] - D'[i]|$. $\qquad\square$

*Proof of Lemma 4.*

$$
\begin{aligned}
\|c_D - c_{D'}\|_1 &= \int_r \int_c |\sum_{i \in n}(I_{D_i \in [c,c+r]} - I_{D'_i \in [c,c+r]})| \\
&\leq \int_r \int_c \sum_{i \in n} |(I_{D_i \in [c,c+r]} - I_{D'_i \in [c,c+r]})| \\
&= \sum_{i \in n} \int_r \int_c |(I_{D_i \in [c,c+r]} - I_{D'_i \in [c,c+r]})| \\
&= 2 \sum_{i \in n} \int_r \min\{\frac{\epsilon}{n}, r\} \\
&\leq 2 \sum_{i \in n} \int_r \frac{\epsilon}{n} \\
&= 2\epsilon
\end{aligned}
$$

$\qquad\square$

*Proof of Lemma 5.* We first state the following lemma, whose proof is deferred to the end.

**Lemma 13.** *Consider two 1-d datasets $\boldsymbol{D}$ and $\boldsymbol{D}'$ over points $p_1, ..., p_k$ s.t. the points in $\boldsymbol{D}$ are each repeated $z_1, ..., z_k$ times for $k \in [n]$ and $0 \leq z_i \leq n$, and the points $\boldsymbol{D}'$ are each repeated $z'_1, ..., z'_k$ times for $k \in [n]$ and $0 \leq z'_i \leq n$. Let $t = \sum_{i \in [k]} |z_i - z'_i|$. We have that $\|c_D - c_{D'}\|_1 \leq \frac{1}{2}t$.*

In our setting, let $p_1, ..., p_k$ be the distinct element of $\boldsymbol{D}$, let $z_1, ..., z_k$ be the number of times each element is repeated in $\boldsymbol{D}^1$ and $z'_1, ..., z'_k$ be the number of times each element is repeated in $\boldsymbol{D}^2$. By Lemma 13, we have that $\|c_{D^1} - c_{D^2}\|_1 \leq \frac{1}{2} \sum_{i \in [k]} |z_i - z'_i|$. Now observe that $z_i = \sum_{j \in [n]} \boldsymbol{m}^1[j] I_{\boldsymbol{D}[j]=p_i}$ and similarly $z'_i = \sum_{j \in [n]} \boldsymbol{m}^2[j] I_{\boldsymbol{D}[j]=p_i}$. Thus, we have

$$
\begin{aligned}
\|c_{D^1} - c_{D^2}\|_1 &\leq \frac{1}{2} \sum_{i \in [k]} |z_i - z'_i| \\
&= \frac{1}{2} \sum_{i \in [k]} |\sum_{j \in [n]} I_{\boldsymbol{D}[j]=p_i}(\boldsymbol{m}^1[j] - \boldsymbol{m}^2[j])| \\
&\leq \frac{1}{2} \sum_{i \in [k]} \sum_{j \in [n]} I_{\boldsymbol{D}[j]=p_i} |\boldsymbol{m}^1[j] - \boldsymbol{m}^2[j]| \\
&= \frac{1}{2} \sum_{j \in [n]} |\boldsymbol{m}^1[j] - \boldsymbol{m}^2[j]|.
\end{aligned}
$$

$\qquad\square$

*Proof of Lemma 6.* For simplicity of notation, we prove this lemma with $\bar{D}$ renamed as $\boldsymbol{D}$ and $\bar{D}'$ renamed as $\boldsymbol{D}'$. For any fixed $r$, we first bound $\|c_D(., r) - c_{D'}(., r)\|_1$. Let $z = r - \lfloor \frac{r}{\frac{1}{u}} \rfloor$ (recall that $u = 2\lceil \frac{k}{2\epsilon} \rceil - 2$). Let $i$ be the index of the first element in which $\boldsymbol{D}$ and $\boldsymbol{D}'$ differ, and let $p = \min\{\boldsymbol{D}[i], \boldsymbol{D}'[i]\}$. Such an index exists as the datasets are different (also recall that any difference is repeated $k$ times, by construction). Therefore, the functions are identical in the range $[-1, p - r)$. Now consider two cases, when $r > \frac{1}{u}$, and when $r \leq \frac{1}{u}$.

In the first case consider the range $[p - r, p - r + z]$ and $[p - r + z, p - r + \frac{1}{u}]$. Over both ranges the functions are constant, and the functions may only change at $p - r + z$. Since the functions are identical before $p - r$ but change at $p - r$, and one changes more than the other, the difference between the two functions in the range $[p - r, p - r + z]$ is at least $k$, that is $|c_D(c, r) - c_{D'}(c, r)| \geq k$ for $q \in [p - r, p - r + z]$. Now, observe that $p - r + z$ is a multiple of $\frac{1}{u}$, and that it is strictly less than $p$. Thus, all points in $D$ and $D'$ at $p - r + z$ are identical. Therefore, both functions have an identical change at $p - r + z$ so that their difference remains the same. Thus, we have that $|c_D(q, r) - c_{D'}(c, r)| \geq k$ for $q \in [p - r, p - r + \frac{1}{u}]$. Thus, in this case $\|c_D(., r) - c_{D'}(., r)\|_1 \geq k \times \frac{1}{u} > \frac{\epsilon}{k}$ for $u = 2\lceil \frac{k}{2\epsilon} \rceil - 2$.

In the second case, consider the range $[p - r, p]$. Both functions are identical right before at $p - r$, while one changes by at least $k$ more than the other at $p - r$, and the functions are constant over $(p - r, p)$. Therefore, $|c_D(c, r) - c_{D'}(c, r)| \geq k$ for $q \in [p - r, p]$. Thus, in this case $\|c_D(., r) - c_{D'}(., r)\|_1 \geq k \times r$.

Now we have

$$\|c_D - c_{D'}\|_1 = \int_r \int_c |c_D(c, r) - c_{D'}(c, r)|$$
$$= \int_{r < \frac{\epsilon}{k}} \|c_D(., r) - c_{D'}(., r)\|_1 + \int_{r \geq \frac{\epsilon}{k}} \|c_D(., r) - c_{D'}(., r)\|_1$$
$$> k\left(\int_{r < \frac{\epsilon}{k}} r + \int_{r \geq \frac{\epsilon}{k}} \frac{\epsilon}{k}\right)$$
$$= k\left(\frac{(\frac{\epsilon}{k})^2}{2} + (\frac{\epsilon}{k})(1 - \frac{\epsilon}{k})\right)$$
$$= k\left(\frac{\epsilon}{k} - \frac{(\frac{\epsilon}{k})^2}{2}\right)$$

Note that, for $\epsilon < k$, $(\frac{\epsilon}{k})^2 \leq \frac{\epsilon}{k}$ so that

$$\frac{\epsilon}{k} - \frac{(\frac{\epsilon}{k})^2}{2} \geq \frac{\epsilon}{k} - \frac{\frac{\epsilon}{k}}{2} = \frac{\frac{\epsilon}{k}}{2}.$$

As such, we have $\|c_D - c_{D'}\|_1 \geq \frac{\epsilon}{2}$. $\square$

*Proof of Lemma 7.* Recall that points in $D$ and $\hat{D}$ have identical last dimensions.

Define $\mathcal{I}_{q,p}^i$ (similar to proof for Theorem 3 Part (i)) as an indicator function equal to 1 if a record $p$ matches the first $i$ dimensions in $q = (c_1, ..., c_d, r_1, ..., r_d)$, that is if $c_j \leq p_j \leq c_j + r_j$ for all $j \in [i]$, and let $D^* = \{D[i, d] \; \forall i, \mathcal{I}_{q, D[i]}^{d-1} = 1\}$ be a 1-dimensional dataset of the $d$-th attribute of the records of $D$ that matches the first $d - 1$ dimensions of $q$, and similarly define $\hat{D}^* = \{\hat{D}[i, d] \; \forall i, \mathcal{I}_{q, \hat{D}[i]}^{d-1} = 1\}$ for $\hat{D}$. Note that the $d$-th dimension of $\hat{D}^*$ and $D^*$ are identical, so that they both contain a subset of elements of the $d$-th dimension of $D$, where the subset is selected based on the indicator function . Thus, we apply Lemma 5 to $\hat{D}^*$ and $D^*$ (the masks in the lemma are induced based on the indicator functions, i.e., $m^1 = (\mathcal{I}_{q,\hat{D}[1]}^{d-1}, ..., \mathcal{I}_{q,\hat{D}[n]}^{d-1})$ and $m^2 = (\mathcal{I}_{q,D[1]}^{d-1}, ..., \mathcal{I}_{q,D[n]^{d-1}})$) to obtain

$$\|c_{D^*} - c_{\hat{D}^*}\|_1 \leq \frac{1}{2} \sum_{i \in [n]} |\mathcal{I}_{q, D[i]}^{d-1} - \mathcal{I}_{q, \hat{D}[i]}^{d-1}|.$$

Moreover, by definition,

$$\|c_D(c_1, ..., c_{d-1}, ., r_1, ..., r_{d-1}, .) - c_{D'}(c_1, ..., c_{d-1}, ., r_1, ..., r_{d-1}, .)\|_1 = \|c_{D^*} - c_{\hat{D}^*}\|_1.$$

Combining the last two inequalities, we have

$$\|c_D - c_{\hat{D}}\|_1 \le \int_{c_1} \cdots \int_{c_{d-1}} \int_{r_1} \cdots \int_{r_{d-1}} \sum_{i \in [n]} \frac{1}{2} |\mathcal{I}_{q,D[i]}^{d-1} - \mathcal{I}_{q,\hat{D}[i]}^{d-1}|$$

$$\le \frac{1}{2} \sum_{i \in [n]} \int_{c_1} \cdots \int_{c_{d-1}} \int_{r_1} \cdots \int_{r_{d-1}} |\mathcal{I}_{q,D[i]}^{d-1} - \mathcal{I}_{q,\hat{D}[i]}^{d-1}|.$$

Now observe that for any $q$ and $i$

$$|\mathcal{I}_{q,D[i]}^{d-1} - \mathcal{I}_{q,\hat{D}[i]}^{d-1}| \le \sum_{j \in [d-1]} |I_{c_j \le D[i,j] \le c_j + r_j} - I_{c_j \le \hat{D}[i,j] \le c_j + r_j}|,$$

Where $I$ is the indicator function, so that

$$\|c_D - c_{\hat{D}}\|_1 \le \frac{1}{2} \sum_{j \in [d-1]} \sum_{i \in [n]} \int_{c_1} \cdots \int_{c_{d-1}} \int_{r_1} \cdots \int_{r_{d-1}} |I_{c_j \le D[i,j] \le c_j + r_j} - I_{c_j \le \hat{D}[i,j] \le c_j + r_j}|.$$

Note that for any $j$, $|I_{c_j \le D[i,j] \le c_j + r_j} - I_{c_j \le \hat{D}[i,j] \le c_j + r_j}| = 1$ only in the two following scenario:
(1) if $\hat{D}[i,j] \le c_j \le D[i,j]$ and $D[i,j] - c_j \le r_j \le 1$ or (2) $\hat{D}[i,j] - 1 \le c_j \le \hat{D}[i,j]$ and $\hat{D}[i,j] - c_j \le r_j \le D[i,j] - c_j$. For the first scenario, we have that

$$\int_{\hat{D}[i,j]}^{D[i,j]} \int_{D[i,j]-c_j}^{1} 1 \, dr_j \, dc_j = \int_{\hat{D}[i,j]}^{D[i,j]} (1 - D[i,j] + c_j) \, dc_j$$

$$= [(1 - D[i,j])c_j + \frac{c_j^2}{2}]_{\hat{D}[i,j]}^{D[i,j]} \, dc_j$$

$$= -[(1 - D[i,j])\hat{D}[i,j] + \frac{\hat{D}[i,j]^2}{2}] + [(1 - D[i,j])D[i,j] + \frac{D[i,j^2]}{2}]$$

$$= -\hat{D}[i,j] + D[i,j]\hat{D}[i,j] - \frac{\hat{D}[i,j]^2}{2} + D[i,j] - \frac{D[i,j]^2}{2}$$

$$= (D[i,j] - \hat{D}[i,j]) - \frac{1}{2}(D[i,j] - \hat{D}[i,j])^2$$

$$\le \frac{\epsilon}{n}$$

And for the second scenario we have

$$\int_{\hat{D}[i,j]-1}^{\hat{D}[i,j]} \int_{\hat{D}[i,j]-c_j}^{D[i,j]-c_j} 1 \, dr_j \, dc_j = \int_{\hat{D}[i,j]-1}^{\hat{D}[i,j]} (\hat{D}[i,j] - D[i,j]) \, dc_j$$

$$= (\hat{D}[i,j] - D[i,j])$$

$$\le \frac{\epsilon}{n},$$

Thus, we have

$$\|c_D - c_{\hat{D}}\|_1 \le \frac{1}{2} \sum_{j \in [d-1]} \sum_{i \in [n]} \int_{c_1} \cdots \int_{c_{d-1}} \int_{r_1} \cdots \int_{r_{d-1}} |I_{c_j \le D[i,j] \le c_j + r_j} - I_{c_j \le \hat{D}[i,j] \le c_j + r_j}|$$

$$\le \frac{1}{2} \sum_{j \in [d-1]} \sum_{i \in [n]} \frac{2\epsilon}{n}$$

$$= (d-1)\epsilon.$$

$\square$

*Proof of Lemma 8.*

$$\|s_D - s_{D'}\|_1 = \int_{\boldsymbol{q} \in Q_s} |\sum_{i \in [n]} \mathcal{I}_{\boldsymbol{D}_i, \boldsymbol{q}}(\boldsymbol{D}[i, d+1] - \boldsymbol{D}'[i, d+1])|$$

$$\leq \int_{\boldsymbol{q} \in Q_s} \sum_{i \in [n]} \mathcal{I}_{\boldsymbol{D}_i, \boldsymbol{q}} |\boldsymbol{D}[i, d+1] - \boldsymbol{D}'[i, d+1]|$$

$$\leq \int_{\boldsymbol{q} \in Q_s} \sum_{i \in [n]} \mathcal{I}_{\boldsymbol{D}_i, \boldsymbol{q}} \frac{\epsilon}{n}$$

$$\leq \sum_{i \in [n]} \frac{\epsilon}{n} \int_{\boldsymbol{q} \in Q_s} 1$$

$$= \epsilon$$

$\square$

*Proof of Lemma 9.*

$$\|s_D - s_{D'}\|_1 = \int_r \int_c |\sum_{i \in n} (I_{\boldsymbol{D}[i,1] \in [c, c+r]} \boldsymbol{D}[i, 2] - I_{\boldsymbol{D}'[i,1]_i \in [c, c+r]} \boldsymbol{D}[i, 2])|$$

$$\leq \int_r \int_c \sum_{i \in n} |I_{\boldsymbol{D}[i,1] \in [c, c+r]} - I_{\boldsymbol{D}'[i,1] \in [c, c+r]}| \boldsymbol{D}[i, 2]$$

$$\leq \int_r \int_c \sum_{i \in n} |I_{\boldsymbol{D}[i,1] \in [c, c+r]} - I_{\boldsymbol{D}'[i,1] \in [c, c+r]}|$$

$$= \sum_{i \in n} \int_r \int_c |I_{\boldsymbol{D}[i,1] \in [c, c+r]} - I_{\boldsymbol{D}'[i,1] \in [c, c+r]}|$$

$$\leq 2 \sum_{i \in n} \int_r \min\{\frac{\epsilon}{n}, r\}$$

$$\leq 2 \sum_{i \in n} \int_r \frac{\epsilon}{n}$$

$$= 2\epsilon$$

$\square$

*Proof of Lemma 10.* We first state the following lemma, whose proof is deferred to the end.

**Lemma 14.** *Consider two 2-dimensional datasets $\boldsymbol{D}$ and $\boldsymbol{D}'$ over points $\boldsymbol{p}_1, ..., \boldsymbol{p}_k$ s.t. the points in $\boldsymbol{D}$ are each repeated $z_1, ..., z_k$ times for $k \in [n]$ and $0 \leq z_i \leq n$, and the points $\boldsymbol{D}'$ are each repeated $z_1', ..., z_k'$ times for $k \in [n]$ and $0 \leq z_i' \leq n$. Let $t = \sum_{i \in [k]} |z_i - z_i'|$. We have that $\|s_D - s_{D'}\|_1 \leq \frac{1}{2}t$.*

In our setting, let $\boldsymbol{p}_1, ..., \boldsymbol{p}_k$ be the distinct element of $\boldsymbol{D}$, let $z_1, ..., z_k$ be the number of times each element is repeated in $\boldsymbol{D}^1$ and $z_1', ..., z_k'$ be the number of times each element is repeated in $\boldsymbol{D}^2$. By Lemma 14, we have that $\|s_{D^1} - s_{D^2}\|_1 \leq \frac{1}{2} \sum_{i \in [k]} |z_i - z_i'|$. Now observe that $z_i =$

$\sum_{j \in [n]} \boldsymbol{m}^1[j] I_{\boldsymbol{D}[j]=p_i}$ and similarly $z_i' = \sum_{j \in [n]} \boldsymbol{m}^2[j] I_{\boldsymbol{D}[j]=p_i}$. Thus, we have

$$\|s_{D^1} - s_{D^2}\|_1 \leq \frac{1}{2} \sum_{i \in [k]} |z_i - z_i'|$$

$$= \frac{1}{2} \sum_{i \in [k]} |\sum_{j \in [n]} I_{\boldsymbol{D}[j]=p_i}(\boldsymbol{m}^1[j] - \boldsymbol{m}^2[j])|$$

$$\leq \frac{1}{2} \sum_{i \in [k]} \sum_{j \in [n]} I_{\boldsymbol{D}[j]=p_i}|\boldsymbol{m}^1[j] - \boldsymbol{m}^2[j]|$$

$$= \frac{1}{2} \sum_{j \in [n]} |\boldsymbol{m}^1[j] - \boldsymbol{m}^2[j]|.$$

$\square$

*Proof of Lemma 11.* For notational consistency with proof of Theorem 3, we rename $\boldsymbol{D}'$ as $\boldsymbol{D}$ in the proof here.

Consider the dataset $\hat{\boldsymbol{D}}$ that $\hat{\boldsymbol{D}}[i, j] = \bar{\boldsymbol{D}}[i, j]$ for $j \in [d-1]$ and $\hat{\boldsymbol{D}}[i, d] = \boldsymbol{D}[i, d]$. That is $\hat{\boldsymbol{D}}$ is a dataset with its first $d-1$ dimensions the same as $\bar{\boldsymbol{D}}$ and its $d$-th dimension the same as $\boldsymbol{D}$. We have that

$$\|s_D - s_{\bar{D}}\|_1 \leq \|s_D - s_{\hat{D}}\|_1 + \|s_{\hat{D}} - s_{\bar{D}}\|_1$$

We study the first two terms separately. Consider the second term $\|s_{\hat{D}} - s_{\bar{D}}\|_1$. Observe that the first $d-1$ dimensions of $\bar{\boldsymbol{D}}$ and $\hat{\boldsymbol{D}}$ are identical, so that applying any predicate on the first $d-1$ attributes of the datasets leads to the same filtered data points. Furthermore, the $d$-th dimension of the points differ by at most $\frac{\epsilon}{n}$, so by Lemma 9, we have

$$\|s_{\hat{D}} - s_{\bar{D}}\|_1 \leq 2\epsilon.$$

Next, consider the term $\|s_D - s_{\hat{D}}\|_1$. Define $\mathcal{I}_{\boldsymbol{q},\boldsymbol{p}}^i$ an indicator function equal to 1 if a record $\boldsymbol{p}$ matches the first $i$ dimensions in $\boldsymbol{q} = (c_1, ..., c_d, r_1, ..., r_d)$, that is if $c_j \leq p_j \leq c_j + r_j$ for all $j \in [i]$, and let $\boldsymbol{D}^* = \{\boldsymbol{D}[i, d : d+1] \; \forall i, \mathcal{I}_{\boldsymbol{q},\boldsymbol{D}[i]}^{d-1} = 1\}$ be a 2-dimensional dataset of the $d$-th and $d+1$-th attribute of the records of $\boldsymbol{D}$ that matches the first $d-1$ dimensions of $\boldsymbol{q}$, and similarly define $\hat{\boldsymbol{D}}^* = \{\hat{\boldsymbol{D}}[i, d : d+1] \; \forall i, \mathcal{I}_{\boldsymbol{q},\hat{\boldsymbol{D}}[i]}^{d-1} = 1\}$ for $\hat{\boldsymbol{D}}$. Note that the $d$-th and $d+1$-th dimension of $\hat{\boldsymbol{D}}^*$ and $\boldsymbol{D}^*$ are identical, so that they both contain a subset of elements of the $d$-th and $d+1$-th dimension of $\boldsymbol{D}$, where the subset is selected based on the indicator function $\mathcal{I}_{\boldsymbol{q},\boldsymbol{q}}^{d-1}$. Thus, we apply Lemma 10 to $\hat{\boldsymbol{D}}^*$ and $\boldsymbol{D}^*$ (the masks in the lemma are induced based on the indicator functions, i.e., $\boldsymbol{m}^1 = (\mathcal{I}_{\boldsymbol{q},\hat{\boldsymbol{D}}[1]}^{d-1}, ..., \mathcal{I}_{\boldsymbol{q},\hat{\boldsymbol{D}}[n]}^{d-1})$ and $\boldsymbol{m}^2 = (\mathcal{I}_{\boldsymbol{q},\boldsymbol{D}[1]}^{d-1}, ..., \mathcal{I}_{\boldsymbol{q},\boldsymbol{D}[n]}^{d-1})$) to obtain

$$\|s_{D^*} - s_{\hat{D}^*}\|_1 \leq \frac{1}{2} \sum_{i \in [n]} |\mathcal{I}_{\boldsymbol{q},\boldsymbol{D}[i]}^{d-1} - \mathcal{I}_{\boldsymbol{q},\hat{\boldsymbol{D}}[i]}^{d-1}|.$$

Moreover, by definition,

$$\|s_D(c_1, ..., c_{d-1}, ., r_1, ..., r_{d-1}, .) - s_{D'}(c_1, ..., c_{d-1}, ., r_1, ..., r_{d-1}, .)\|_1 = \|s_{D^*} - s_{\hat{D}^*}\|_1.$$

Combining the last two inequalities, we have

$$\|s_D - s_{\hat{D}}\|_1 \leq \int_{c_1} \cdots \int_{c_{d-1}} \int_{r_1} \cdots \int_{r_{d-1}} \sum_{i \in [n]} \frac{1}{2} |\mathcal{I}_{\boldsymbol{q},\boldsymbol{D}[i]}^{d-1} - \mathcal{I}_{\boldsymbol{q},\hat{\boldsymbol{D}}[i]}^{d-1}|$$

$$\leq \frac{1}{2} \sum_{i \in [n]} \int_{c_1} \cdots \int_{c_{d-1}} \int_{r_1} \cdots \int_{r_{d-1}} |\mathcal{I}_{\boldsymbol{q},\boldsymbol{D}[i]}^{d-1} - \mathcal{I}_{\boldsymbol{q},\hat{\boldsymbol{D}}[i]}^{d-1}|$$

$$\leq (d-1)\epsilon$$

Where the last inequality was shown in the proof of Theorem 3. Putting everything together, we have that

$$\|s_D - s_{\bar{D}}\|_1 \leq 2\epsilon + (d-1)\epsilon = (d+1)\epsilon.$$

$\square$

*Proof of Lemma 12.* We have $\|r_D - r_{D'}\|_\mu = \int_0^1 |r_D(q) - r_{D'}(q)| d\mu$. Let $I_{i,q}$ be an indicator variable denoting whether $q \in [\boldsymbol{D}[i], \boldsymbol{D}'[i])$ (or $[\boldsymbol{D}'[i], \boldsymbol{D}[i])$ if $\boldsymbol{D}'[i] < \boldsymbol{D}[i]$). We have that $|r_D(q) - r_{D'}(q)| = \sum_{i=0}^n I_{i,q}$. Thus, we have $\|r_D - r_{D'}\|_\mu = \int_0^1 \sum_{i=0}^n I_{i,q} d\mu = \sum_{i=0}^n \int_0^1 I_{i,q} d\mu = \sum_{i=0}^n \int_{p_{i,1}}^{p_{i,2}} d\mu = \sum_i \mu([\boldsymbol{D}[i], \boldsymbol{D}'[i]])$. $\qquad\square$

*Proof of Lemma 13.* Let $D^* = D \cup D'$, that is $D^*$ is a dataset over points $p_1, ..., p_k$ each repeated $z_1^*, ..., z_k^*$ with $z_i^* = \max\{z_i, z_i'\}$. We have $\|c_D - c_{D'}\| \le \|c_D - c_{D^*}\|_1 + \|c_{D'} - c_{D^*}\|_1$.

By Lemma 15 (stated and proved below) we have

$$\|c_D - c_{D^*}\|_1 \le \frac{1}{2} \sum_{i \in [k]} (z_i^* - z_i) = \frac{1}{2} \sum_{i \in [k]} (\max\{z_i, z_i'\} - z_i)$$

and similarly

$$\|c_{D'} - c_{D^*}\|_1 \le \frac{1}{2} \sum_{i \in [k]} (z_i^* - z_i') = \frac{1}{2} \sum_{i \in [k]} (\max\{z_i, z_i'\} - z_i')$$

so that

$$\|c_D - c_{D'}\|_1 \le \frac{1}{2} \sum_{i \in [k]} (\max\{z_i, z_i'\} - z_i') + \frac{1}{2} \sum_{i \in [k]} (\max\{z_i, z_i'\} - z_i)$$

$$= \frac{1}{2} \sum_{i \in [k]} (\max\{z_i, z_i'\} - z_i + \max\{z_i, z_i'\} - z_i')$$

$$= \frac{1}{2} \sum_{i \in [k]} |z_i - z_i'|$$

$\qquad\square$

*Proof of Lemma 14.* Let $\boldsymbol{D}^* = \boldsymbol{D} \cup \boldsymbol{D}'$, that is $\boldsymbol{D}^*$ is a dataset over points $\boldsymbol{p}_1, ..., \boldsymbol{p}_k$ each repeated $z_1^*, ..., z_k^*$ with $z_i^* = \max\{z_i, z_i'\}$. We have $\|s_D - s_{D'}\| \le \|s_D - s_{D^*}\|_1 + \|s_{D'} - s_{D^*}\|_1$.

By Lemma 16 (stated and proved below) we have

$$\|s_D - s_{D^*}\|_1 \le \frac{1}{2} \sum_{i \in [k]} (z_i^* - z_i) = \frac{1}{2} \sum_{i \in [k]} (\max\{z_i, z_i'\} - z_i)$$

and similarly

$$\|s_{D'} - s_{D^*}\|_1 \le \frac{1}{2} \sum_{i \in [k]} (z_i^* - z_i') = \frac{1}{2} \sum_{i \in [k]} (\max\{z_i, z_i'\} - z_i')$$

so that

$$\|s_D - s_{D'}\|_1 \le \frac{1}{2} \sum_{i \in [k]} (\max\{z_i, z_i'\} - z_i') + \frac{1}{2} \sum_{i \in [k]} (\max\{z_i, z_i'\} - z_i)$$

$$= \frac{1}{2} \sum_{i \in [k]} (\max\{z_i, z_i'\} - z_i + \max\{z_i, z_i'\} - z_i')$$

$$= \frac{1}{2} \sum_{i \in [k]} |z_i - z_i'|$$

$\qquad\square$

**Lemma 15.** *Consider a 1-dimensional database $D$ of size $n$, such that points $p_1, ..., p_k$ in $D$ are each repeated $z_1, ..., z_k$ times for $k, z_i \in [n]$, $\sum_{i \in [n]} z_i = n$. Consider another dataset $D'$ containing the a subset of points $p_1, ...p_k$, having $0 \le z_i' \le z_i$ and $\sum_{i \in [n]} z_i' = n'$ so that $n' \le n$. Let $t = \sum_{i \in [k]} |z_i - z_i'|$. We have that $\|c_D - c_{D'}\|_1 = t \int_{r=0}^1 r = \frac{1}{2} t$.*

*Proof of Lemma 15.* Fix a value for $r$. Let $X = \{i \in [k], z_i \ne z_i'\}$.

Observe that $c_D$ and $c_{D'}$ only differ for queries where there exists an $i \in X$ for which $c \in [p_i - r, p_i]$. Thus,

$$\|c_D - c_{D'}\| = \int_{r=0}^{1} \int_{q=-1}^{1} \sum_{i \in [k]} I_{p_i \in (q,r)}(z_i - z_i')$$

$$= \int_{r=0}^{1} \sum_{i \in [k]} (z_i - z_i') \int_{q=-1}^{1} I_{p \in (q,r)}$$

$$= \int_{r=0}^{1} \sum_{p \in [k]} (z_i - z_i') \int_{q=p_i-r}^{p_i} 1$$

$$= \int_{r=0}^{1} rt = \frac{t}{2}$$

.                                                                                                  □

**Lemma 16.** *Consider a 2-dimensional database $D$ of size $n$, such that points $p_1, ..., p_k$ in $D$ are each repeated $z_1, ..., z_k$ times for $k, z_i \in [n]$, $\sum_{i \in [n]} z_i = n$. Consider another dataset $D'$ containing the a subset of points $p_1, ...p_k$, having $0 \le z_i' \le z_i$ and $\sum_{i \in [n]} z_i' = n'$ so that $n' \le n$. Let $t = \sum_{i \in [k]} |z_i - z_i'|$. We have that $\|s_D - s_{D'}\|_1 = t \int_{r=0}^{1} r = \frac{1}{2} t$.*

*Proof of Lemma 16.* Fix a value for $r$. Let $X = \{i \in [k], z_i \ne z_i'\}$.

Observe that $s_D$ and $s_{D'}$ only differ for queries where there exists an $i \in X$ for which $c \in [p_i[1] - r, p_i[1]]$. Thus,

$$\|s_D - s_{D'}\| = \int_{r=0}^{1} \int_{c=-1}^{1} \sum_{i \in [k]} I_{p_i[1] \in (c,r)}(z_i - z_i')p_i[2]$$

$$\le \int_{r=0}^{1} \sum_{i \in [k]} (z_i - z_i') \int_{c=-1}^{1} I_{p_i[1] \in (c,r)}$$

$$= \int_{r=0}^{1} \sum_{i \in [k]} (z_i - z_i') \int_{c=p_i[1]-r}^{p_i[1]} 1$$

$$= \int_{r=0}^{1} rt = \frac{t}{2}$$

.                                                                                                  □

