# OpenReview forum: "Towards Establishing Guaranteed Error for Learned Database Operations"
_ICLR.cc/2024/Conference — ICLR 2024 poster_

### Official Review · Reviewer_Px8G · 2023-10-30

**Soundness:** 2 fair
**Presentation:** 3 good
**Contribution:** 2 fair
**Rating:** 3
**Confidence:** 3

**Summary:**

This paper works on establishing the lower bound of the required model size for arbitrary datasets, given a tolerable error parameter. The authors provide worst-case and average-case theoretical analysis for three database operations, i.e., learned index, learned cardinality estimation, and range-sum estimation. Some empirical evaluations are performed. While this appears to be the first theoretical study of such guarantees for the learned model, there are several concerns.

**Strengths:**

1. The paper is the first to study the guaranteed error for learned database operations.
2. This paper is overall easy to read, though I did not read the proof.

**Weaknesses:**

1. Concerns about the problem setting -- the studied problem does not quite correspond to the learned DB operators
-- For the learned index, it is fine for the model to make errors as we can maintain some delta to help correct the error. After all, the training and testing data are the same.
-- For the learned cardinality estimator, the absolute error is not what people are using for evaluation -- q-error is. This is because for a true cardinality of 10K, an error of 100 is more acceptable than an error of 10 for a true cardinality of 10.

2. When people study learned db operators, the main advantage is that the model can adapt to the underlying data and hence derive instance-optimal solutions. However, this paper is studying the worst-case/average scenario, which contradicts to the motivation of using learned model to be instance-optimal.

3. It would be great to show more implications of this theoretical results and how can we make use of this theoretical results in practice. It is now unclear. The result shown in Figure 1 does not provide actional items for users. In particular, it seems that different models are having similar average-case lower bounds in terms of error -- which is not quite differentiable.

**Questions:**

1. It would be great to better articulate the problem settings (See W1)
2. It would be great to justify the usefulness of worst-case/average-case guarantees instead of instance-level guarantees (See W2).
3. It would be great to show more implications of the theoretical results (see W3)
4. what is the unique property of learned DB operators that are used during the proof? Putting it in another way, is this method applicable to arbitrary functions and not limited to learned DB operators?

---

> ### Author Response · Authors · 2023-11-16
>
> We thank the reviewer for the comments. We have updated the paper based on the reviewers’ comments, and changes are marked in blue in the updated file.
>
> R4Q1/W1. Regarding the error metric for learned indexing, we are unsure what the reviewer means by “maintaining delta”. The typical practice in learned indexing is to do binary (or exponential) search (Kraska et al., 2018; Ferragina & Vinciguerra, 2020; Ding et al., 2020) from the final estimate on the sorted array to find the exact answer, in which case, as R3Q3 mentions one may be interested in the $\log_2$ error of the model (which is the runtime of this binary search step). We have added a discussion to the newly added Appendix A showing how our bounds can be translated to log2 error, as discussed next. Ideally, to ensure $\log_2$ error is at most an error parameter $\tau$, setting $\epsilon=2^\tau$ in our bounds would provide the required model size given the $\log_2$ error $\tau$. This is true when considering worst-case error (i.e., Theorem 1) and in our upper-bounds on the required model size when considering average case error (i.e., Theorem 4, Lemma 2 and part (ii) of Theorems 2,3,5). That is, our bounds on the mentioned results simply translate to bounds for $\log_2$ error by replacing $\epsilon$ with $2^\tau$. However, this is not true in the case of lower bounds for average error, (i.e., Corollary 1 and part (i) of Theorems 2,3,5) and the required model size can be smaller than what the results would suggest if we set $\epsilon=2^\tau$. We leave study of the log2 error in these cases to future work, although we expect similar proof techniques that were used in this paper to be applicable to that case as well.
>
>
> Regarding the error for cardinality estimation, although q-error is a useful metric for the evaluation of cardinality estimators, worst-case guarantees on absolute error can provide more meaningful guarantees on the performance of a system. For example, Negi et al. 2021 show that a cardinality estimator that achieves a worst-case q-error of 4100 can lead to a similar runtime as another method with a worst-case q-error of 100. Indeed a worst-case bound on q-error is a worse indicator of the system’s performance compared with a worst-case bound on absolute error. For instance, the q-error of 4100 can correspond to a query with true cardinality 4100 and estimated cardinality 1 (in which case one needs to read an extra 4100 items from disk, which will take little resources, both computation and memory wise) while the error can also correspond to an estimate of 1,000,000 for a query whose true cardinality is 4,100,000,000 (in which case one needs to read an extra 4 billion records, a huge computation and memory cost). On the other hand, the absolute error in those two cases will respectively be 4,399 and  4,099,000,000 a much better indicator of the extra computation caused by the error of the cardinality estimator. Overall, a worst-case bound on absolute error can readily translate to performance metrics (e.g., worst-case memory consumption and number of disk accesses), while worse-case q-error in itself is not informative.
>
> Finally, note that cardinality estimation is not just used for query optimization, and has use-cases in approximate query processing (similar to range-sum estimation), where the goal is to return to the user an estimated query answer (e.g., for the query of how many people are in an area), which is common in analytical applications. In such settings, the absolute error is useful as it provides a bound on what the true query answer can be, and is often directly shown to the user as error bars.

---

> > ### Author Response · Authors · 2023-11-16
> >
> > R4Q3/W3. Please see the newly added paragraph to the introduction and our response R2Q2 for a discussion on the implications of our theoretical results. Here, we describe in more detail two practical use cases enabled by our theoretical results.
> >
> > The first practical use case is to ensure that the model size used is at least what the bounds suggest. This ensures that, as data instances change or new data instances are observed, the model has enough capacity to answer queries and reduces the need for hyperparameter tuning.  For instance, while a linear model (which has a very small size) may be sufficient to answer queries for data from a uniform distribution, if the data changes one will need a more complex model (of larger size). By choosing a model size that guarantees a desired accuracy level across data instances (i.e., using our worst-case guarantees), one can avoid redoing hyperparameter tuning as the data instance changes.
> >
> > Another practical use case of our results is in resource management for database systems, especially significant for cloud service providers that store and manage the learned models for a large number of database instances. Our bounds can tell such service providers how much resources they need to be able to support learned models in their systems with a desired accuracy level. This can be used in cost calculation, and to make sure enough resources are available to support a learned model across all instances (i.e., to provide reliability guarantees).
> >
> > R4Q2/W2. We would like to clarify that in the context of the instance-optimal framework of learned models, our bounds study the worst-case instances for such instance-optimal methods. That is, we study how high the error of a model can be on an instance, even if we optimize it for that instance. Such worst-case guarantees are useful when one needs to reason about the performance of a system without having seen the specific data instances it will be used on. Two examples of when this is useful are presented in our response to R4Q2. We present further details here.
> >
> > Worst-case guarantees serve a different purpose than instance-level guarantees. Instance-level guarantees will have to depend on the properties of the instance (e.g., bounds in Zeighami & Shahabi, 2023 depend on the p.d.f of data distribution while bounds in Ferragina et al., 2020 depend on the distribution of gaps in an array). Such bounds can only be calculated after the instance is observed, and will need to be recomputed if an instance changes. This does not allow a system to provide apriori guarantees on the performance of the system, which worst-case guarantees provide. For instance, a cloud service provider needs to know how much resources it needs to allocate to learned models apriori (e.g., to calculate costs and ensure availability). Instance level guarantees do not help, since instances will change and so will the amount of resources needed. However, a worst-case guarantee allows the service provider to calculate the maximum amount of resources it needs, and therefore guarantee the reliability of the system. A similar example holds in terms of setting model size, where one may not be able to afford to redo hyperparameter tuning when the data instance changes, and thus setting model size to what is suitable across instances will be beneficial, which can be done through worst-case bounds.

---

> > > ### Author Response · Authors · 2023-11-16
> > >
> > > R4Q4.  Our results only apply to database operations and not arbitrary functions. We study the properties of database functions to provide bounds on the required model size to represent database operations. Our proofs specifically study the properties of the three database operations in question (i.e., indexing, cardinality estimation and range-sum estimation). The main property of database operations used in our proofs is: if two databases differ in a set of points S, how different the answer to queries will be on those two databases? Our proofs utilize properties of the set S to bound the difference between query answers on different databases. Such bounds are query-dependent, and our proofs study this for the three different database operations (presented in technical Lemmas 3-8 in our appendix, which are one of our main technical contributions). The consequence of such a bound is that if a model (with a specific parameter setting, e.g., after training) achieves a small error on some dataset, then it will also achieve a small error on other datasets that don’t differ from it too much, but will achieve large error on datasets that do differ from it too much (the bound is used to rigorously quantify this statement). Thus, we can count how many different model parameter settings are needed to achieve small errors on all possible datasets, which directly gives a bound on the required model size (as we need to use enough bits to realize all the required different model parameter settings). Please also see the newly added section Appendix B1 and our response to R3Q2 for more intuition on the proofs.

---

### Official Review · Reviewer_EAu4 · 2023-11-01

**Soundness:** 3 good
**Presentation:** 2 fair
**Contribution:** 4 excellent
**Rating:** 8
**Confidence:** 2

**Summary:**

The authors prove lower bounds on model sizes for various learned database components (indexing, cardinality estimation, and range-sum estimation) for a given maximum error and domain size across any dataset. For example, using the author's theorems, one can compute a lower bound on the number of bits a learned index structure must use to achieve a worst-case error over all databases of N rows and domain size U. The authors also give results for average case error, assuming either a uniform or an arbitrary query distribution.

**Strengths:**

This paper tackles an important problem. Existing learned index structures either grow unbounded to support a specific error (e.g., PGM index), or have an unbounded error but a specific size (e.g., RMI). In the former case, the author's bounds can be used to estimate the size of the fixed-error index structure ahead of time. In the latter case, where the model size is fixed ahead of time and the error is determined during training, the author's bounds can be used to estimate an initial model size from the desired error, or to compute an estimate of the error from the size of the model. Interestingly, the authors show that the domain size is relevant for worse case behavior but not for average case behavior.

**Weaknesses:**

While the bounds given by the authors certainly help bring our understanding of learned database components closer to that of traditional data structures, it is not clear to me how these bounds could be used in systems today. The most I seem to be able to say with the author's bounds is "if your learned index uses S bytes of memory, then for a given domain size, there exists a dataset size n for which your index must have an error larger than e."

It is not clear to me how to use these bounds to size a particular structure with a fixed error, which would be useful for estimating the size of error-bounded indexes. Nor is it clear to me how to use these bounds to estimate the error of a fixed-size structure. Further, the authors fall short of actually *constructing* either (1) a dataset of size `n` and domain size `u` for which a particular index cannot achieve an error better than `e` for a given size, or (2) a index structure that can actually achieve the given error bounds at the specified size.

That said, this paper is still a useful contribution for practitioners like me! Using the worst-case error bounds for indexing given in table 1, setting n=200M, u=2^32, e = 8, I get sizes remarkably close to the exhaustive search performed by the cited learned index papers (7MB). So, even if these bounds are not exactly what database folks need, they appear to be a useful heuristic.

**Questions:**

Q1) I found the main text of the paper to be both overly formal and to give very little intuition about the actual proof. If the main text is just going to go over the results and implications, you might as well drop the formal notation and give intuitive / illustrated examples of each problem and bound.

Q2) That said, I really do wish the authors had tried harder to convey the intuition behind the proof in the main text. Unfortunately, I do not have the required background to parse the appendices. As far as I can tell, the authors assume that a learned structure can, at most, represent 2^n different functions of bitstrings. The authors show an isomorphism between bitstring functions and learned indexing, and then assert that, since only 2^n bitstring functions can be covered, any database admitting more than 2^n values must have at least imperfect prediction. From there, the authors work through to the error bound. I would love to know more!

Q3) A note on error functions: learned index structures today generally care about log-loss (i.e., the average log2 of the absolute differences), since this is the number of binary search steps one will need for the "last mile" search.

Q4) typo A.1 "on the notions of the notions of"

---

> ### Author Response · Authors · 2023-11-16
>
> We thank the reviewer for the comments. We have updated the paper based on the reviewers’ comments, and major changes are marked in blue in the updated file.
>
> R3, Response to comment “ it is not clear to me how these bounds could be used in systems … error bounds at the specified size. “.
> Please see our response to R2Q2 and R4Q3 regarding practical use cases of our bounds.
>
> R3Q1-2. We have added a new subsection to Appendix B, further discussing the intuition behind our proofs which we present here. As, the reviewer observed, given that there are $2^\sigma$ possible bitstrings (for model size $\sigma$), multiple datasets must be mapped to the same model parameter values (that is, after training, multiple datasets will have the exact same model parameter values). To obtain a bound on the error, the proof shows that some datasets that are mapped to the same parameter values will be too different from each other (i.e., queries on the datasets have answers that are too different), so that the exact same parameter setting cannot lead to answering queries accurately for both datasets. The proofs do this by constructing $2^\sigma+1$ different datasets such that query answers differ by more than $2\epsilon$ on all the $2^\sigma+1$ datasets. Thus two of those datasets must be mapped to the same model parameter values, and the model must have an error more than $\epsilon$ on at least one of them, which completes the proof. The majority of the proofs are on how this set of $2^\sigma+1$ datasets is constructed. Specifically, the proofs construct a set of datasets, where each pair differs in some set of points $S$. The main technical challenge is constructing/showing that for any of the two datasets that differ in a set of points $S$, the maximum or average difference between the query answers is at least $2\epsilon$. This last statement is query dependent, and our technical Lemmas 3-8 in the appendix are proven to quantify the difference between query answers between the datasets based on the properties of the set $S$ and the query type. This is especially difficult in the average-case scenario, as one needs to study how the set $S$ affects all possible queries.
>
> R3Q3. We have added a discussion to the newly added Appendix A showing how our bounds can be translated to $\log_2$ error, as discussed next. Ideally, to ensure $\log_2$ error is at most an error parameter $\tau$, setting $\epsilon=2^\tau$ in our bounds would provide the required model size given the $\log_2$ error $\tau$. This is true when considering worst-case error (i.e., Theorem 1) and in our upper bounds on the required model size when considering average case error (i.e., Theorem 4, Lemma 2 and part (ii) of Theorems 2,3,5). That is, our bounds on the mentioned results simply translate to bounds for $\log_2$ error by replacing $\epsilon$ with $2^\tau$. However, this is not true in the case of lower bounds for average error, (i.e., Corollary 1 and part (i) of Theorems 2,3,5) and the required model size can be smaller than what the results would suggest if we set $\epsilon=2^\tau$. We leave the study of the $\log_2$ error in these cases to future work, although we expect similar proof techniques that were used in this paper to be applicable to that case as well.

---

### Official Review · Reviewer_q6av · 2023-11-01

**Soundness:** 2 fair
**Presentation:** 3 good
**Contribution:** 2 fair
**Rating:** 5
**Confidence:** 4

**Summary:**

The work investigates the minimal size of models (minimal number of bits needed to represent model parameters) that can approximate ranks and (weighted) orthogonal range counts with a guaranteed maximal additive error (epsilon). It derives formulas for the worst/average-case model size and compares them with empirical results of a limited empirical result.

**Strengths:**

S1) Theoretical results are accompanied by empirical results.

S2) Results provide some insight into the practical complexity of approximating some database operators for multidimensional numerical data with learned models.

S3) The paper is generally easy to read and understand.

**Weaknesses:**

W1) Presentation a bit misleading: The paper gives the impression as if the results apply to general database operators over all sorts of tabular data (e.g., SQL queries over mix of categorial/numerical data) while the results are limited to orthogonal/axis-aligned range queries (intersection of range selections along each dimensions). In general the limitations of this work are not outlined clearly.

W2) Significance unclear: The empirical study is too limited to give a clear idea how much predictive power is gained via the derived formulas and the general discussion does not clearly explain the implications for learned indexing and related topics. Non-learned baselines such as random sampling are missing.

W3) Literature: The related work discussion is too limited in scope. It does not discuss VC dimensionality of orthogonal range queries (which is well-known and discussed in the referenced works), epsilon approximations in the computational geometry literature and data summaries in the database literature that pose questions that are similar in spirit to the model size question just with non-learned models such as samples and histogram-based data structures. Examples of related work in the aforementioned topics:

- Mustafa, N. H., & Varadarajan, K. R. (2017). Epsilon-approximations and epsilon-nets. arXiv preprint arXiv:1702.03676.

- Suri, S., Tóth, C. D., & Zhou, Y. (2004, June). Range counting over multidimensional data streams. In Proceedings of the twentieth annual symposium on Computational geometry (pp. 160-169).

- Shekelyan, M., Dignös, A., & Gamper, J. (2017). Digithist: a histogram-based data summary with tight error bounds. Proceedings of the VLDB Endowment, 10(11), 1514-1525.

- Cormode, G., Garofalakis, M., Haas, P. J., & Jermaine, C. (2011). Synopses for massive data: Samples, histograms, wavelets, sketches. Foundations and Trends® in Databases, 4(1–3), 1-294.

- Wei, Z., & Yi, K. (2018). Tight space bounds for two-dimensional approximate range counting. ACM Transactions on Algorithms (TALG), 14(2), 1-17.

W4) Minor issues

- p.3, Preliminaries, "specifics" => "specifies"
- p.6, Learned Cardinality Estimation, "estiamtion" => "estimation"
- p.10, Related Work, "hyperparametr" => "hyperparameter"

**Questions:**

Q1) What are the limitations of this work?

Q2) What is a non-trivial prediction that is enabled by the results in this this work?

Q3) What topics in the literature are related to the studied topic and how do the results in this work relate to the results from the related topics (e.g., how much better is a learned model than random sampling / other summaries)?

---

> ### Author Response · Authors · 2023-11-16
>
> We thank the reviewer for the comments. We have updated the paper based on the reviewers’ comments, and major changes are marked in blue in the updated file.
>
> R2W1/Q1: The reviewer has correctly pointed out that our results on cardinality and range-sum estimation only apply to axis-parallel range queries for numerical data (which is a very common setting, e.g., Kipf et al., 2018; Ma & Triantafillou, 2019; Zeighami et al. 2023). We have modified the introduction to specify this earlier on. We have also added Sec. A to the appendix to discuss various other considerations not addressed in the paper, such as cardinality estimation for joins, other aggregation functions and other error metrics.
>
> R2W2/Q2: We have added a new paragraph to the introduction further clarifying the implication of our results, as discussed next. There are two important predictive statements made by our theorems, following two interpretations of our theorems. First, our results provide a lower bound on the worst-case error given a model size (this view is presented in our experiments). This shows what error is achievable by a model of a certain size. Our experiments illustrate that our bound on error is meaningful, showing that models achieve error values close to what the bound suggests. Second, our results provide a lower bound on the required model size to achieve the desired accuracy level across datasets. This has important implications for resource management in database systems. For instance, this helps a cloud service provider decide how much resources it needs to allocate (and calculate the cost) for the learned model to be able to guarantee an accuracy level. Moreover, our bound provides the first theoretical guideline on how the model size should be set in practice to be able to guarantee a desired accuracy level.
>
> Moreover, based on the reviewer’s suggestions, we have added random sampling as a baseline in Figure 1. The figure shows that sampling performs worse than learned models for uniform distribution while it performs similarly for GMMs (Sample should be compared with Linear as they both have the same size). We note that although presenting results for sampling in our experiments provides more context to our empirical study, the goal of our experiments is not a detailed empirical evaluation of different modeling choices, and we refer the reviewer to the related work (e.g., Kipf et al., 2018; Zeighami et al. 2023) for such a comparison. The goal of our experiments is to show our theoretical bounds are meaningful, showing that existing modeling choices achieve similar error values as our bound suggests. We also note that our theoretical bounds suggest (discussed in Sec. 5 and also in our response to R2W3) that the gap between learned models and sampling will grow as dimensionality increases (we use one-dimensional datasets in our experiments).
>
> R2W3/Q3. We thank the reviewer for pointing out the related work and the interesting connection between our results and non-learned data models. We have updated the related work section based on the reviewer’s suggestion. Please see the newly added paragraph in the related work section for a detailed answer to the reviewer’s question. To summarize, the newly added paragraph includes a discussion of theoretical results in the related non-learned literature (such as $\epsilon$-approximation, sampling and use of VC dimensionality, non-learned data summaries). We specifically compare our bounds with bounds on $\epsilon$-approximations, showing that $\epsilon$-approximations (which include random sampling) have a worse lower bound compared with our theorems in high dimensions. This shows that random sampling (and other $\epsilon$-approximations) can be a much worse data model in terms of space efficiency compared with other modeling choices in high dimensions.

---

> > ### Comment · Reviewer_q6av · 2023-11-22
> >
> > Thank you that clears up my questions!
> >
> > Note: Looking at some other answers I do share the concern as to how this applies to query optimisation, i.e., multiplicative (q-error) errors are usually considered due to the error propagation and if the bounds relate only to the final precision (after error propagation) then they it would not tell us anything meaningful about models before error propagation. It would seem to make a big difference if the trained model is just for one combination of attributes (one particular join) or for any possible combination of attributes (adhoc joins) when there's too many attributes to prepare models for all combinations (typical problem).

---

> > > ### Author Response · Authors · 2023-11-23
> > >
> > > First, we would like to clarify that earned models indeed support “adhoc join operations” (see Kipf et al., 2018; Negi et al., 2023), Consequently, our theoretical analysis appropriately takes into account the support for adhoc join operations. Second, it's essential to underscore that, given the inherent capability of learned models to facilitate arbitrary joins between tables, there is no necessity to factor in error propagation. This is because when using learned models one does not need to combine answers from different estimates, and can use a single model forward pass to estimate the cardinality for an operation in the query plan. As a result, our bound on absolute error can be used to provide error guarantees for operations within a query plan, which in turn translates to performance guarantees on the total cost of a query plan. We elaborate on these points further below.
> > >
> > > First, we note that error propagation, as discussed in Sec. 3.1 of [a] (which discusses how error of multiple cardinality estimates can be combined throughout a join), does not apply to learned models. Learned models are trained to provide estimates of any part of a query plan without having to combine estimates from multiple inferences. That is, learned models are trained to support various adhoc joins, as done in Kipf et al., 2018; Negi et al., 2023, so that a single model is trained for various join operations and the same model is used to estimate the cardinality of different joins. For example, one can use the same learned model to first estimate the cost of a join between $R_1\bowtie R_2$, as well as $R_3\bowtie R_4$, and perform another model inference to obtain the cardinality of  $(R_1\bowtie R_2)\bowtie (R_3\bowtie R_4)$. The cardinality estimate of $(R_1\bowtie R_2)\bowtie (R_3\bowtie R_4)$ is not calculated by combining the estimates for $R_1\bowtie R_2$ and $R_3\bowtie R_4$, but instead by a single separate model inference. Thus, an absolute error bound on the model error, provides a bound on *both* intermediary cardinality estimates (throughout the query plan) and the final cardinality estimate (i.e., estimate of answer size).
> > >
> > > Therefore, such a bound can be used to provide performance guarantees for a query plan. For instance, if the cardinality estimate for some join operation in the plan is $k$ and the absolute error bound of the model is $\epsilon$, then the output of this (possibly intermediate)  join operation requires memory no more than $\epsilon + k$. Furthermore, one can similarly calculate an upper bound on the runtime of each operation by bounding the size of any intermediary step. That is, if two tables have cardinality estimates $k_1$ and $k_2$, and assuming join of two tables of size $s_1$ and $s_2$ takes time $T(s_1, s_2)$ (for some function $T$ which depends on the join algorithm used, see [a] for $T$ corresponding to merge sort join), one can bound the time to perform the join as $T(k_1+\epsilon, k_2+\epsilon)$.  As mentioned earlier, since the errors don’t accumulate (and each cardinality estimate is a result of a new model forward pass), one can always bound the cardinality of any intermediary step with estimate $k$ as $k+\epsilon$. This, in turn, provides an upper bound for the total cost of any operation within a query plan, which, in turn, can be used to provide an upper bound on the total cost of a query plan. To summarize, a bound on the absolute error can be used to provide a bound on the cost of a given plan. This is useful, as one can calculate how much extra computational and memory resources will be used due to the error of the cardinality estimation model, and set the model size accordingly.
> > >
> > > Finally, as mentioned above, in existing work (Kipf et al., 2018; Negi et al., 2023) a single learned model is trained to support various joins across different relations. Here, we discuss how our bounds apply to such a scenario. Indeed, our discussion in Appendix A and response to R1W1 applies in the case of arbitrary joins. Since we consider worst-case bounds, we can use the worst-case of possible joins to provide a bound on the required model size. That is, we can apply our bound to the case of the join with the largest output size across all tables. Among typical joins in a relational database this can be a full outer join of all tables. Thus, we can use our bounds by setting $n=n_J$ and $d=d_1+ …+ d_k$, where $n_J$ is the size of the full outer join of all $k$ tables in the database, and $d_1, …, d_k$ are the dimensionality of each table.
> > >
> > > [a] Moerkotte, Guido, Thomas Neumann, and Gabriele Steidl. "Preventing bad plans by bounding the impact of cardinality estimation errors." Proceedings of the VLDB Endowment 2.1 (2009): 982-993.

---

> > > > ### Comment · Reviewer_q6av · 2023-11-23
> > > >
> > > > Thanks, that's a an interesting comment and may help to clear up some misunderstandings.

---

### Official Review · Reviewer_Aapp · 2023-11-03

**Soundness:** 3 good
**Presentation:** 3 good
**Contribution:** 3 good
**Rating:** 8
**Confidence:** 3

**Summary:**

In this paper, the authors consider the problem of providing
guarantees for learned database operations. In particular, they
consider indexing, cardinality estimation, and range-sum estimation,
and provide error guarantees for these three operations in both the
worst and average case (in the latter, they consider uniform
distribution on queries, and a second scenario where any distribution
can be used). Besides, they provide an experimental evaluation
comparing their error bounds with the errors obtained by training
different models on datasets sampled from different distributions.

**Strengths:**

* As pointed out by the authors, for database operations like
  indexing, learned estimators have empirically been shown to
  outperform some well-known traditional methods. However, such
  learned estimators are not widely used as no guarantees on their
  errors are known. In this sense, this paper makes a significant
  contribution by providing such guarantees for some useful database
  operations.

* The experimental evaluation provides some evidence that the bounds
  provided in the paper are meaningful.

* The paper is well written, with clear statements of the problems studied and the results obtained.

**Weaknesses:**

* The cardinality estimation queries considered in the paper are
  restrictive. For such queries to be useful in practical database
  systems, they should include more complex queries, in particular,
  the join operator. In fact, one of the most important cardinality
  estimation tasks in databases is the estimation of the size of a
  join query, which is witnessed by the large number of research
  articles on this subject.

* The range-sum estimation queries considered in the paper are also
  restrictive. In particular, including other forms of aggregation
  would be very useful for practical database applications.

**Questions:**

* Could you please comment on the two points mentioned in Weaknesses.
  In particular, could you please comment on the possibility of using
  the ideas of the paper to provide error guarantees for aggregate
  operators min, max, and average.

---

> ### Author Response · Authors · 2023-11-16
>
> We thank the reviewer for the comments. We have updated the paper based on the reviewers’ comments, and major changes are marked in blue in the updated file.
>
> R1W1. We have added a paragraph in the newly added Appendix A discussing how our results may apply to joins as follows.  A naive extension of our bounds to the cardinality estimation for joins is to apply the bound to the join of tables. That is, if two tables have respectively $d_1$ and $d_2$ dimensions and if their join consists of $n_J$ elements, then we can apply our bounds in Table 1 with $d=d_1+d_2$ and $n=n_J$ to obtain a lower bound on the required model size for estimating the cardinality of the join. However, we expect such an approach to overestimate the required model size, as it does not consider the join relationship between the two tables. For instance, $n_J\times d$ may be much larger than $n_1\times d_1+n_2\times d_2$, because of duplicate records created due to the join operations. Considering the join relationship, one may be able to provide bounds that depend on the original table sizes and not the join size.
>
> R1W2. We have added a paragraph in the newly added Appendix A that discusses providing bounds for other aggregation functions as follows.  Although we expect similar proof techniques as what was used to apply to other aggregations such as min/max/avg, we note that studying worst-case bounds for min/max/avg may not be very informative. Intuitively, this is because, for such aggregations, one can create arbitrary difficult queries that require the model to memorize all the data points. For instance, consider datasets for which the $(d+1)$-th dimension, where the aggregation is applied, only takes 0 or 1 values, while the other dimensions can take any values in a domain of size $u$. Now avg/min/max for any range query will have an answer between 0 and 1 (for min/max the answer be exactly 0 or 1). However, unless the model memorizes all the points exactly (which requires a model size close to the data size), its worst-case error can be up to 0.5. This is because queries with a very small range can be constructed that match exactly one point in the database, and unless the model knows the exact value of all the points, it will not be able to provide a correct answer for all such queries. Note that the error of 0.5 is very large for avg/min/max queries, and a model that always predicts 0.5 also obtains a worst-case error 0.5. This is not the case for sum/count aggregations whose query answers range between 0 to $n$, and a model with an absolute error of 0.5 can be considered a very good model for most queries when answering sum/count queries.
>
> To summarize, worst-case errors of min/avg/max queries are disproportionately affected by smaller ranges, while smaller ranges have a limited impact on the worst-case error of count/sum queries.  As such we expect that for the case of min/max/avg queries, one needs to study the error for a special set of queries (e.g., queries with a lower bound on their range, as done in  Zeighami et al. (2023), which also makes similar observations) to be able to obtain meaningful bounds. We leave such a study to future work.

---

### Meta-Review · Area_Chair_J3R8 · 2023-12-04

**Metareview:**

This paper presents new ideas for learned database operations.  While the reviewers did not come to consensus, several felt that the results are interesting.  The theory can be practically demonstrated and it paves the way for future learned data structures/operations in databases. I

**Justification For Why Not Higher Score:**

The paper could have looked at more range of subproblems and further developed the area.  I may consider a spotlight, but I think a poster is sufficient.

**Justification For Why Not Lower Score:**

The paper formalizes ideas on using predictions in databases.  This has generally been of interest, but there has been a lack of theory shown.  This has potential to be influential.

---

### Decision · Program_Chairs · 2024-01-16

Accept (poster)